# A unique intracellular tyrosine in neuroligin-1 regulates AMPA receptor recruitment during synapse differentiation and potentiation

Mathieu Letellier[1,2], Zsófia Szíber[1,2], Ingrid Chamma[1,2], Camille Saphy[1,2], Ioanna Papasideri[1,2], Béatrice Tessier[1,2], Matthieu Sainlos[1,2], Katalin Czöndör[1,2] & Olivier Thoumine[1,2]

To better understand the molecular mechanisms by which early neuronal connections mature into synapses, we examined the impact of neuroligin-1 (Nlg1) phosphorylation on synapse differentiation, focusing on a unique intracellular tyrosine (Y782), which differentially regulates Nlg1 binding to PSD-95 and gephyrin. By expressing Nlg1 point mutants (Y782A/F) in hippocampal neurons, we show using imaging and electrophysiology that Y782 modulates the recruitment of functional AMPA receptors (AMPARs). Nlg1-Y782F impaired both dendritic spine formation and AMPAR diffusional trapping, but not NMDA receptor recruitment, revealing the assembly of silent synapses. Furthermore, replacing endogenous Nlg1 with either Nlg1-Y782A or -Y782F in CA1 hippocampal neurons impaired long-term potentiation (LTP), demonstrating a critical role of AMPAR synaptic retention. Screening of tyrosine kinases combined with pharmacological inhibitors point to Trk family members as major regulators of endogenous Nlg1 phosphorylation and synaptogenic function. Thus, Nlg1 tyrosine phosphorylation signaling is a critical event in excitatory synapse differentiation and LTP.

[1] Interdisciplinary Institute for Neuroscience, UMR 5297, Univ. Bordeaux, F-33000 Bordeaux, France. [2] Interdisciplinary Institute for Neuroscience, UMR 5297, CNRS, F-33000 Bordeaux, France. These authors contributed equally: Zsófia Szíber, Ingrid Chamma. Correspondence and requests for materials should be addressed to M.L. (email: mathieu.letellier@u-bordeaux.fr) or to O.T. (email: olivier.thoumine@u-bordeaux.fr)

The assembly of synapses allowing communication between neurons represents a crucial process in the development of the central nervous system. Maintaining a proper balance between the activity of excitatory and inhibitory synapses is of crucial importance to normal brain function, as alterations of this parameter are associated with several brain disorders such as autism or schizophrenia[1–4]. At excitatory synapses, glutamate release is juxtaposed to glutamate receptors stabilized at the postsynaptic density by scaffolding proteins such as PSD-95 and S-SCAM, whereas at inhibitory synapses GABA is released near GABA receptors interacting with the multimodal scaffolding protein gephyrin[5,6]. However, the molecular mechanisms by which the proper postsynaptic neurotransmitter receptors are precisely assembled in front of the corresponding presynaptic terminals to transmit specific information remain essentially unresolved.

Neuroligins (Nlgs) are postsynaptic, trans-synaptic adhesion molecules that recognize presynaptic neurexins (Nrxs) with high affinity[7]. A number of studies manipulating Nlgs expression levels suggest that these molecules are critically implicated in synapse assembly, possibly through competition mechanisms[8,9]. Specifically, neurons expressing high levels of Nlgs compared to neighboring neurons seem to receive a higher number of synaptic inputs and vice versa[10–14]. Rodents express four Nlgs that are differently distributed at excitatory vs. inhibitory synapses: while Nlg1 is predominant at excitatory synapses, Nlg2 and Nlg4 are specifically found at inhibitory synapses, and Nlg3 is found at both types of synapses[15–17]. In agreement with their subcellular distribution, Nlg isoforms contribute differently to the development and function of glutamatergic vs. GABAergic synapses, through their capacity to assemble appropriate scaffolds and receptors in front of corresponding presynaptic terminals. Indeed, Nlg1 favors the differentiation of glutamatergic synapses by trapping surface-diffusing AMPA receptors (AMPARs) via the PSD-95 scaffold and recruiting NMDA receptors (NMDARs) via its extracellular domain[18–21]. In contrast, Nlg2 recruits $GABA_A$ or glycine receptors through a specific interaction with collybistin, thus bringing gephyrin to the cell membrane and driving the assembly of inhibitory synapses[5,6,15,22,23]. Intriguingly, however, the intracellular tail of Nlg1 comprises both conserved gephyrin- and PDZ domain-binding motifs and binds equally well gephyrin and PSD-95 in vitro[23–25]. In addition, previous studies show that overexpressing or knocking-down Nlg1 in cultured neurons equally affect excitatory and inhibitory synapses number[10–12,26], Therefore, the specific role and distribution of Nlg1 at glutamatergic but not GABAergic synapses remains unclear. One mechanism that may enable the specificity of Nlg1 to glutamatergic synapses involves the phosphorylation of a critical intracellular tyrosine (Y782), which inhibits gephyrin binding, instead promoting PSD-95 recruitment[25].

Yet, many open questions remain regarding this mechanism including: (i) is Nlg1 tyrosine phosphorylation able to control the recruitment of glutamate vs GABA receptors, through the assembly of their respective postsynaptic scaffolds? (ii) If yes, what is the physical mechanism of receptor recruitment and the effects on synaptic transmission and potentiation? (iii) Is there a cross talk with presynaptic differentiation? (iv) What is the effect of Nlg1 tyrosine phosphorylation on the density of dendritic spines? (v) What is the impact of manipulating the phosphorylation level of endogenous Nlg1 through specific tyrosine kinases? To answer those questions, we used a combination of biochemistry, immunocytochemistry, single molecule imaging, pharmacology, and patch-clamp recordings in cultured neurons and brain slices. We demonstrate that phosphorylation of the Nlg1 Y782 regulates the differentiation and long-term potentiation (LTP) of excitatory synapses by primarily affecting the recruitment of AMPARs.

## Results

**Nlg1 tyrosine mutants differentially bind gephyrin in vitro**. We previously showed by in vitro pull-down that 16-aa Nlg1 peptides comprising the gephyrin-binding site bound the purified E-domain of gephyrin, and that introduction of a phosphate group on Y782 abolished binding[25] (Fig. 1a). One mutant peptide (Y782F) exhibited strong gephyrin binding, while another mutant (Y782A) did not bind gephyrin and therefore phenocopies the tyrosine phosphorylation. To examine the impact of Nlg1 tyrosine phosphorylation on the binding of Nlg1 to full length gephyrin or PSD-95, we performed pull-down experiments using recombinant GST proteins fused to the intracellular domain of Nlg1, either from wild-type Nlg1 (Nlg1-WT), or from Nlg1 carrying Y782A or Y782F mutations (Fig. 1b–d). Compared to GST-Nlg1-WT, GST-Nlg1-Y782F strongly bound gephyrin-Venus from COS cell lysates, while GST-Nlg1-Y782A bound as little gephyrin as the negative control GST alone (Fig. 1c). In contrast, all GST-Nlg1 proteins equally pulled down purified PSD-95-mCherry (Fig. 1d), most likely through the conserved C-terminal PDZ domain-binding motif[24]. Thus, mutation of Y782 selectively regulates gephyrin binding in vitro.

**Nlg1 mutants differently associate with scaffolds in neurons**. We next investigated whether Nlg1 point mutants differentially associate with endogenous scaffolds when expressed in primary rat hippocampal neurons. To detect Nlg1, we used a selective biotinylation strategy[27] followed by surface labeling with Atto647-conjugated monomeric streptavidin (mSA), a small probe (~3 nm) that penetrates synapses very well[28]. We used recombinant Nlg1 carrying both extracellular splice sites A and B, since this is the predominant transcript expressed in the hippocampus[29]. Neurons were electroporated with AP-tagged Nlg1 mutants at 0 days in vitro (DIV), together with biotin ligase (BirA[ER]) and GFP as a volume marker, and Nlg1 distribution was examined at DIV 14–15. Biotinylated Nlg1-WT and mutants exhibited a clustered distribution at the membrane, allowing us to quantify the number of Nlg1 clusters containing immunostained gephyrin or PSD-95 (Fig. 1e–h). While the majority of Nlg1-WT clusters associated with PSD-95 (~55%), a few clusters (~20%) associated with gephyrin. Strikingly, the Y782A mutation decreased by ~40% the proportion of Nlg1 clusters containing gephyrin but did not affect the association with PSD-95 (Fig. 1f, h). In contrast, the Y782F mutation decreased by ~20% the amount of Nlg1 clusters containing PSD-95, and slightly increased those containing gephyrin (Fig. 1f, h). To assess whether the mutations also affected the amount of scaffolds within single Nlg1 clusters, we measured the PSD-95 or gephyrin fluorescence intensity. The gephyrin signal in Nlg1 clusters was significantly increased by the Y782F mutation, but not by Y782A (Supplementary Fig. 1a). In contrast, the PSD-95 signal at Nlg1 puncta was not affected by either of the two Nlg1 mutants (Supplementary Fig. 1b). These results suggest that Y782 directly controls the interaction of Nlg1 with gephyrin, and indirectly affects PSD-95 binding.

To assess whether those differences in scaffold recognition were accompanied by changes in the membrane dynamics of Nlg1 mutants, we performed single molecule tracking of biotinylated Nlg1 using sparse Atto594-conjugated mSA labeling[28] (Fig. 1i). Nlg1-Y782A had a similar global diffusion coefficient to Nlg1-WT, while Nlg1-Y782F diffused significantly faster (Fig. 1j, k). This result is consistent with the fact that Nlg1-Y782F binds better to gephyrin scaffolds that are less abundant than PSD-95 puncta (Fig. 1f, h), thus offering more space for extra-scaffold diffusion.

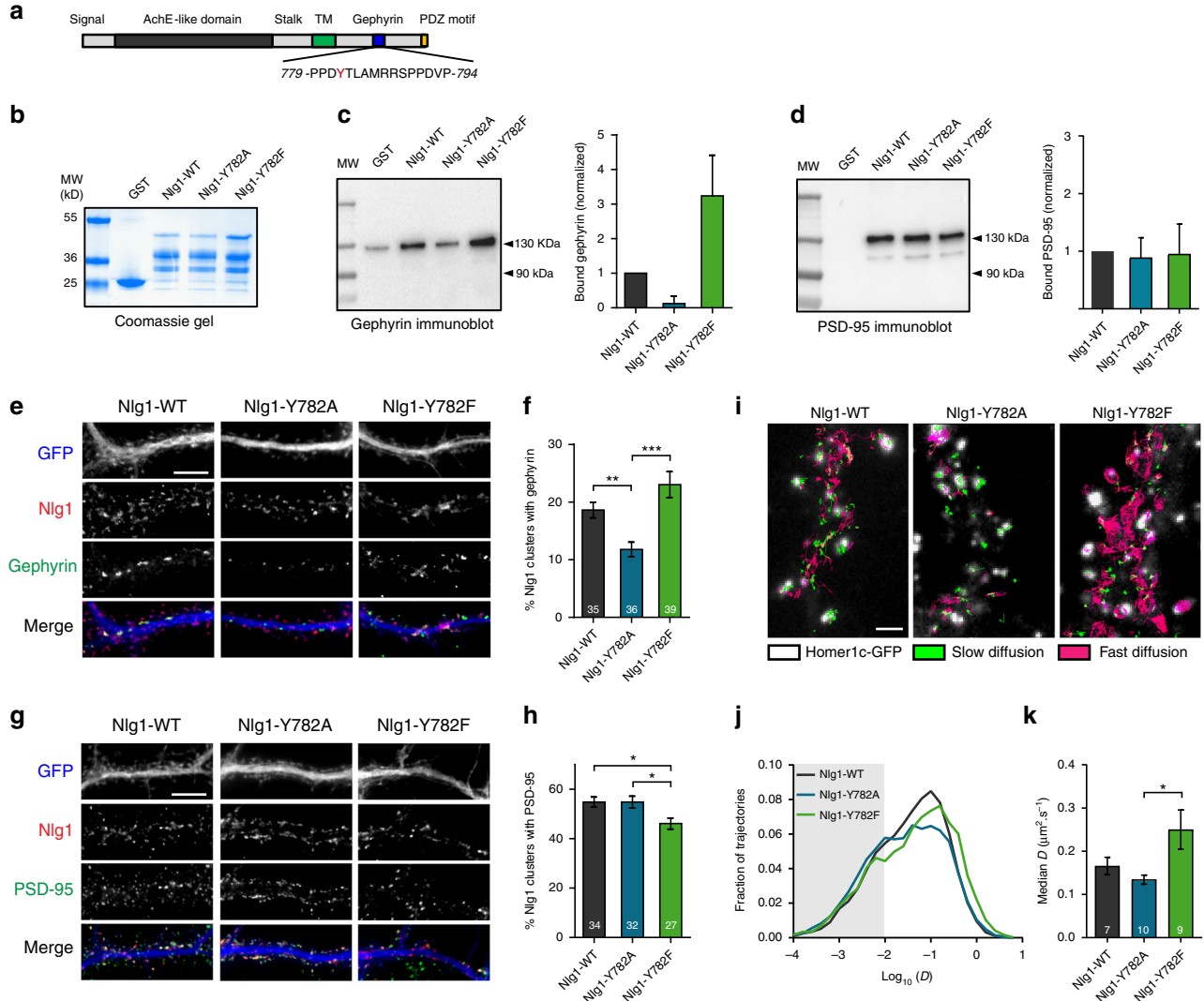

**Fig. 1** Nlg1 mutants differentially associate with gephyrin and PSD-95 scaffolds. **a** Schematics of Nlg1 structure. The sequence of the gephyrin-binding motif (blue) is shown with Y782 in red; AchE: acetylcholine esterase-like domain; TM: transmembrane domain. **b** Coomassie gel loaded with GST or GST-Nlg1 fusion proteins (Nlg1-WT, Y782A, or Y782F). These proteins migrate as three major bands between 25 kD and 50 kD, suggesting that they experience some degradation. The upper bands at 50 kD corresponding to the full length molecule are expected to bind PSD-95. This upper band plus the major band at ~40 kD are expected to bind gephyrin. **c, d** Immunoblots of gephyrin-Venus and PSD-95-mCherry pulled down by GST-Nlg1 proteins, respectively, and corresponding graphs of bound gephyrin and PSD-95, normalized to GST-Nlg1-WT (mean ± SEM of 2 independent experiments). **e, g** Dendrites from DIV 14–15 cultured neurons electroporated at DIV 0 with AP-tagged Nlg1-WT, -Y782A, or -Y782F (red) along with BirA[ER] and GFP (blue). Surface Nlg1 was labeled with Atto647-conjugated mSA (red) and endogenous gephyrin or PSD-95 were immunostained, respectively (green). Scale bars, 10 μm. **f, h** Percentage of surface Nlg1 clusters positive for gephyrin or PSD-95 for the three conditions (number of cells indicated within the bars, from 3 independent experiments). **i** Representative trajectories of individual Nlg1 molecules tracked by uPAINT (see Methods) at the surface of neurons at DIV 14–15 expressing Nlg1-WT, -Y782A, or -Y782F along with BirA[ER] and Homer1c-GFP (white), sparsely labeled with Atto594-conjugated mSA. Magenta and green represent fast ($D > 0.01\ \mu m^2/s$) and slow ($D \le 0.01\ \mu m^2/s$) diffusing molecules, respectively. Scale bar, 2 μm. **j** Distributions of individual diffusion coefficients in log scale (26013, 28756, and 17894 trajectories from 7, 10, and 9 cells for Nlg1-WT, Nlg1-Y782A, and Nlg1-Y782F, respectively). **k** Median diffusion coefficient for the 3 conditions (from one experiment). Data in graphs **f**, **h**, and **k** were compared by a Kruskal–Wallis test followed by Dunn's multiple comparison test (*$P < 0.05$, **$P < 0.01$, ***$P < 0.001$). Data represent mean ± SEM

**Effect of Nlg1 mutants on excitatory synapse differentiation.** We next examined the effects of expressing Nlg1 point mutants on glutamatergic synapse differentiation, by measuring the recruitment of the presynaptic vesicular transporter VGlut1 and postsynaptic AMPARs in DIV 14–15 dissociated hippocampal neurons, using immunocytochemistry. Nlg1-WT expression increased by ~50% the density of VGlut1 and GluA1 clusters compared to control neurons expressing empty vector (EV) (Fig. 2a–c), confirming that Nlg1 expression promotes the formation of excitatory synapses[10,12,26,30]. The increase in VGlut1 cluster density was similar for both Nlg1 mutants, as expected

since they share the same extracellular domain and show similar surface binding to a soluble Nrx1β-Fc ligand (Supplementary Fig. 1c, d). In contrast, the GluA1 cluster density was enhanced upon Nlg1-Y782A expression, but unchanged by the Y782F mutant (Fig. 2a–c). This suggests that the Y782F mutation, which withstands phosphorylation and promotes the interaction of Nlg1 with gephyrin, prevents the recruitment of AMPARs in front of corresponding glutamatergic terminals.

We also examined the impact of Nlg1 point mutants on glutamatergic transmission by performing whole-cell patch-clamp recordings of miniature AMPAR-mediated excitatory

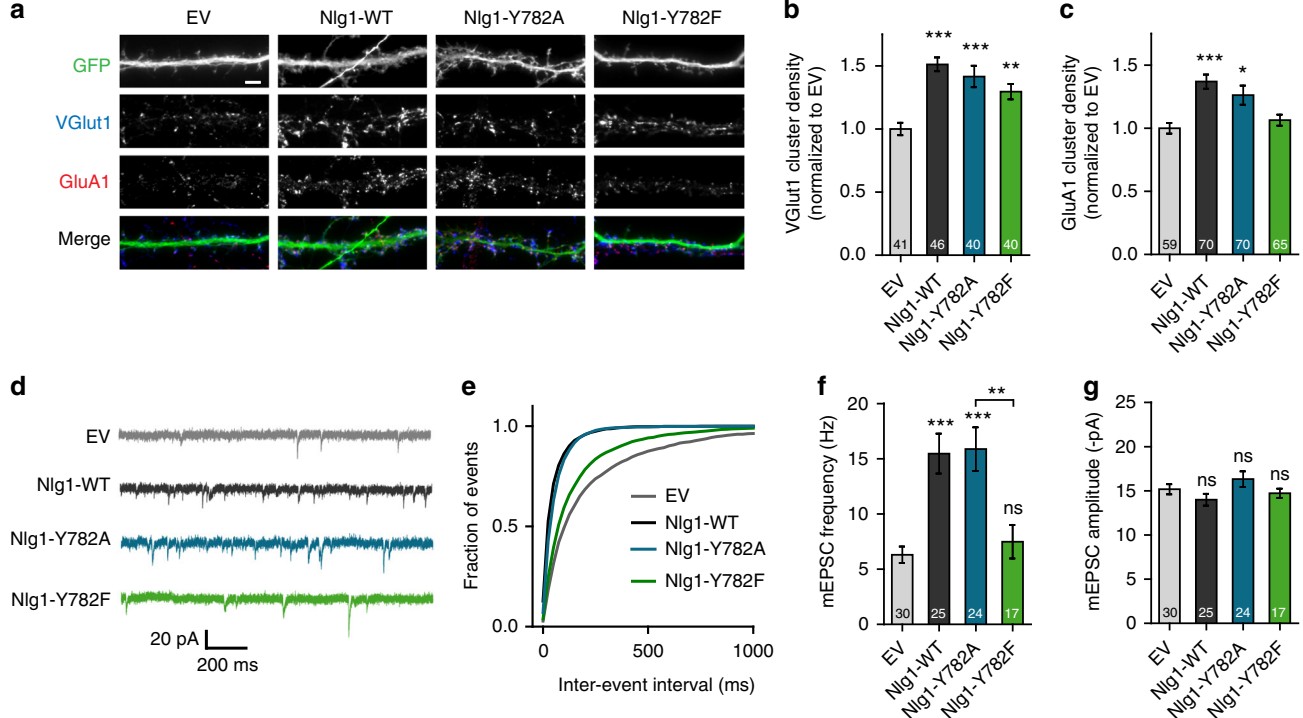

**Fig. 2** Effects of Nlg1 mutants on excitatory synapse differentiation. **a** Dendrites from DIV 14–15 cultured neurons co-transfected with GFP (green) and either EV, Nlg1-WT, -Y782A, or -Y782F. Neurons were live stained with an N-terminal GluA1 antibody (red) and counterstained with a VGlut1 antibody (blue). Scale bar, 5 μm. **b**, **c** Density of VGlut1 or GluA1 clusters, respectively, normalized to the EV condition (number of cells indicated within the bars, from 3 independent experiments). **d** Representative traces of mEPSCs recordings from DIV 14–15 neurons expressing EV, Nlg1-WT, -Y782A, or -Y782F clamped at −70 mV in the presence of TTX and bicuculline. **e** Cumulative distributions of the mEPSCs inter-event intervals for the corresponding conditions. **f**, **g** Mean mEPSC frequencies and amplitudes, respectively, for each condition (number of cells indicated within the bars, from 3 independent experiments). Data in graphs **b**, **c**, **f**, and **g** represent mean ± SEM and were compared by a Kruskal–Wallis test followed by Dunn's multiple comparison test (ns: not significant, *$P < 0.05$, **$P < 0.01$, ***$P < 0.001$)

postsynaptic currents (mEPSCs) (Fig. 2d–g). Compared to EV, Nlg1-WT overexpression tripled the frequency of mEPSCs (Fig. 2d–f)[10], consistent with an increase in the number of glutamatergic inputs formed on transfected cells (Fig. 2b). While Nlg1-Y782A had a similar effect as Nlg1-WT, Nlg1-Y782F prevented the increase in mEPSC frequency (Fig. 2d–f), in agreement with its difficulty to recruit PSD-95 and AMPARs (Fig. 2c). The mEPSC amplitudes were similar for all conditions (Fig. 2g), suggesting that the amount of AMPARs at individual postsynapses, once they are assembled, is relatively stable. Together, these results suggest that Nlg1-Y782A, but not Nlg1-Y782F, promotes the assembly of functional excitatory synapses.

**Nlg1-Y782A but not Y782F trap surface-diffusing AMPARs.**
To investigate the trafficking mechanism underlying the differential recruitment of AMPARs by the two Nlg1 point mutants, we measured AMPAR membrane diffusion. We expressed Nlg1-WT or Nlg1 mutants at DIV 7, together with Homer1c-GFP as a postsynaptic marker[21], and monitored AMPAR surface mobility at DIV 10 using single molecule tracking[28]. To label endogenous AMPARs, we used an antibody against the N-terminal domain of the GluA1 subunit[31], further detected by anti-rabbit Fab conjugated to a photostable organic dye (Atto594) (Fig. 3a). At high Fab density, strong GluA1 labeling was observed at synapses colocalizing with Homer1c-GFP, revealing good probe penetrability in the synaptic cleft (Supplementary Fig. 2a). Furthermore, the GluA1/Fab mix strongly labeled neurons expressing SEP-GluA1, demonstrating antibody specificity (Supplementary

Fig. 2b). At low Fab concentration, we detected the motion of individual AMPARs (Fig. 3b), whose trajectories were analyzed to yield their mean squared displacement (MSD) over time (Fig. 3c), and extract the diffusion coefficient (Fig. 3d). In agreement with previous results using anti-GluA1 coated Quantum dots[21], the global median AMPAR diffusion was decreased by ~60% upon expression of Nlg1-WT, compared to control cells transfected with EV (Fig. 3e). This effect was also observed with Nlg1-Y782A but not Nlg1-Y782F, suggesting that the failure of this Nlg1 mutant to assemble postsynaptic scaffolds (Fig. 3f) makes it unable to capture surface AMPARs. Overall, there was an inverse relationship between AMPAR diffusion and the number of postsynapses, acting as trapping elements[32] (Fig. 3e, f).

**Nlg1 mutants differently affect AMPAR- vs NMDAR EPSCs.**
To examine whether Nlg1 mutants also affected excitatory synaptic transmission in a preparation which preserves network connectivity and dendritic architecture, we expressed AP-tagged Nlg1 mutants together with BirA[ER] and GFP in CA1 pyramidal cells from organotypic hippocampal slices, using single-cell electroporation (Fig. 4a)[28]. We cultured slices from Nlg1 knockout (KO) mice to prevent the potential formation of homodimers between Nlg1 mutants and endogenous Nlg1[20,33]. Staining the slices with fluorescent streptavidin showed that all GFP-positive cells expressed biotinylated Nlg1-WT or -Y782A/F mutants (Supplementary Fig. 3a)[28]. We used a double-patch configuration to record AMPAR- and NMDAR-mediated EPSCs upon stimulation of Schaffer's collaterals, in both GFP-expressing CA1 neurons and neighboring non-electroporated neurons,

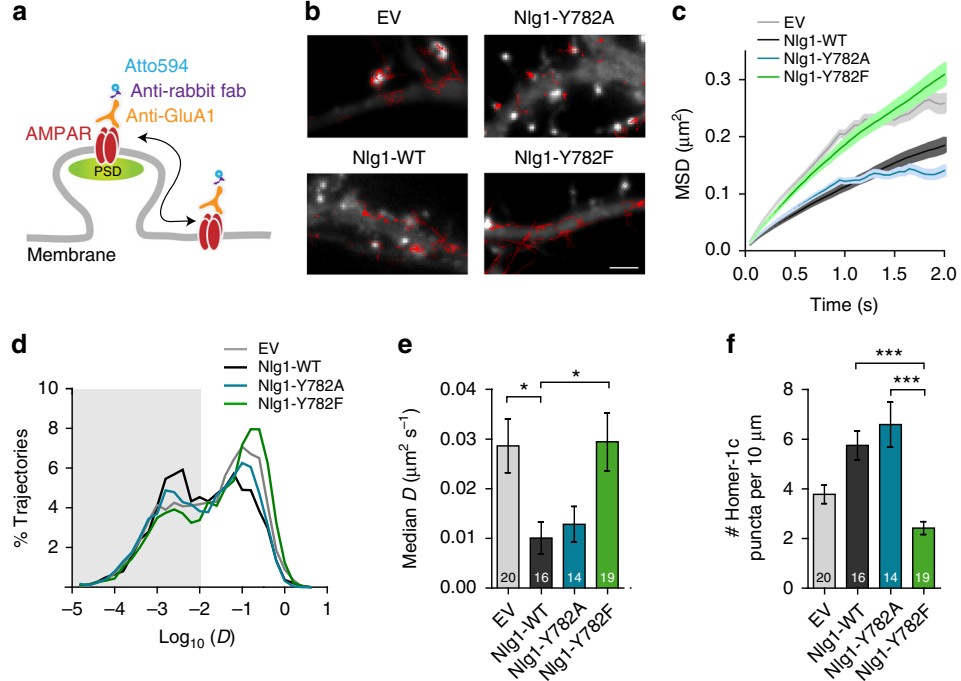

**Fig. 3** Nlg1-Y782A but not –Y782F trap surface-diffusing AMPARs. **a** Neurons were live labeled for endogenous AMPARs with a GluA1 antibody mixed with Atto594-conjugated anti-rabbit Fab. The postsynaptic density (PSD) was detected with Homer1c-GFP. **b** Representative trajectories of individual AMPARs at the neuronal surface (red) in DIV 10 neurons expressing EV, Nlg1-WT, -Y782A, or -Y782F and Homer1c-GFP (white). Scale bar, 2 μm. **c** Average mean square displacement (MSD) and **d** distributions of individual diffusion coefficients in log scale from one experiment, for the four conditions (number of cells/number of trajectories, EV: 8/5690; Nlg1-WT: 7/4340; Nlg1-Y782A: 8/18059; Nlg1-Y782F: 8/16978). Note the higher MSDs and larger peaks of mobile receptors for EV and Nlg1-Y782F, compared to Nlg1-WT and -Y782A. The fraction of slowly mobile molecules on the left of the histogram ($D < 0.01\,\mu m^2\,s^{-1}$) is gray-shaded. **e** The median diffusion coefficient per cell was averaged for each condition (3 independent experiments). The number of cells examined is given in the columns. **f** Effect of the Nlg1 mutations on the number of Homer-1c puncta per unit length of dendrite (3 independent experiments). Data in graphs **e** and **f** represent mean ± SEM and were compared by a Kruskal–Wallis test followed by Dunn's multiple comparison test (*$P < 0.05$, ***$P < 0.001$)

which are contacted by an equivalent number of axons[20]. Amplitudes of EPSCs recorded in cells expressing EV were in the same range as non-electroporated counterparts, validating the dual recording design. In contrast, Nlg1-WT expression induced a robust 4-fold increase of AMPAR-mediated EPSCs compared to paired non-electroporated cells (Fig. 4b, c and Supplementary Fig. 3b), as previously reported[20,34]. Since only the frequency and not the amplitude of AMPA mEPSCs was increased in dissociated neurons (Fig. 2d–g), this effect likely reflects an increase in the number of presynaptic inputs on the Nlg1-expressing cell compared to its non-electroporated neighbor[33]. Moreover, the paired-pulse ratio (PPR) was not significantly changed when expressing Nlg1-WT (Supplementary Fig. 4), suggesting that presynaptic function is not altered. Similarly to Nlg1-WT, Nlg1-Y782A expression shifted AMPAR-mediated EPSCs to higher values compared to neurons expressing EV, whereas Nlg1-Y782F expression did not (Fig. 4b, c and Supplementary Fig. 3b). Nlg1-WT and Nlg1-Y782A also increased the amplitude of spontaneous AMPAR-mediated currents synchronous to those measured in non-electroporated cells, while Nlg1-Y782F had little effect (Fig. 4f, g). Those measurements further show that Nlg1-Y782A, but not Nlg1-Y782F, promotes AMPAR-dependent synaptic transmission.

Because Nlg1 was previously shown to regulate NMDAR-dependent synapse maturation[14,19,26], we also measured the effect of expressing Nlg1 mutants on NMDAR-dependent EPSCs in CA1 neurons from KO slices. Expression of Nlg1-WT increased NMDAR-EPSCs relatively to Nlg1 KO neighbors[26,34,35], and this effect was maintained with the Nlg1-Y782A mutant (Fig. 4b, d and Supplementary Fig. 3c). Surprisingly, Nlg1-Y782F also enhanced NMDAR-EPSC amplitude, resulting in an increased NMDA to AMPA ratio (Fig. 4d, e and Supplementary Fig. 3c). Therefore, despite its inability to recruit AMPARs, Nlg1-Y782F can still recruit NMDARs adjacent to glutamatergic terminals, strongly suggesting that Nlg1 recruits AMPARs and NMDARs through distinct mechanisms.

**Nlg1 C-terminal truncation alters AMPAR-mediated EPSCs.** To better understand how Nlg1 Y782 regulates the selective enhancement of AMPAR- and not NMDAR-mediated synaptic transmission, we used two additional Nlg1 intracellular domain truncation mutants, one construct lacking the last 5-aa comprising the PDZ domain-binding motif (Nlg1-Δ5), and the other construct truncated of the C-terminal 72 aa (Nlg1-Δ72) also including the gephyrin-binding motif (Fig. 5a). In the dual patch-clamp paradigm, both Nlg1-Δ5 and Nlg1-Δ72 significantly prevented the increase of AMPAR-mediated EPSC amplitudes, but not NMDAR-EPSCs, relative to neurons expressing Nlg1-WT (Fig. 5b–d). These results indicate that AMPAR assembly at Nlg1-induced synapses critically depends on the Nlg1 C-terminus, while NMDAR recruitment does not require this motif, relying for instance on a direct coupling to the Nlg1 extracellular domain[19]. Furthermore, the similar behaviors of Nlg1-Y782F and the two Nlg1 truncation mutants support the concept that the phosphorylation of Y782 indirectly regulates the binding of PSD-95 to the PDZ domain-binding motif of Nlg1.

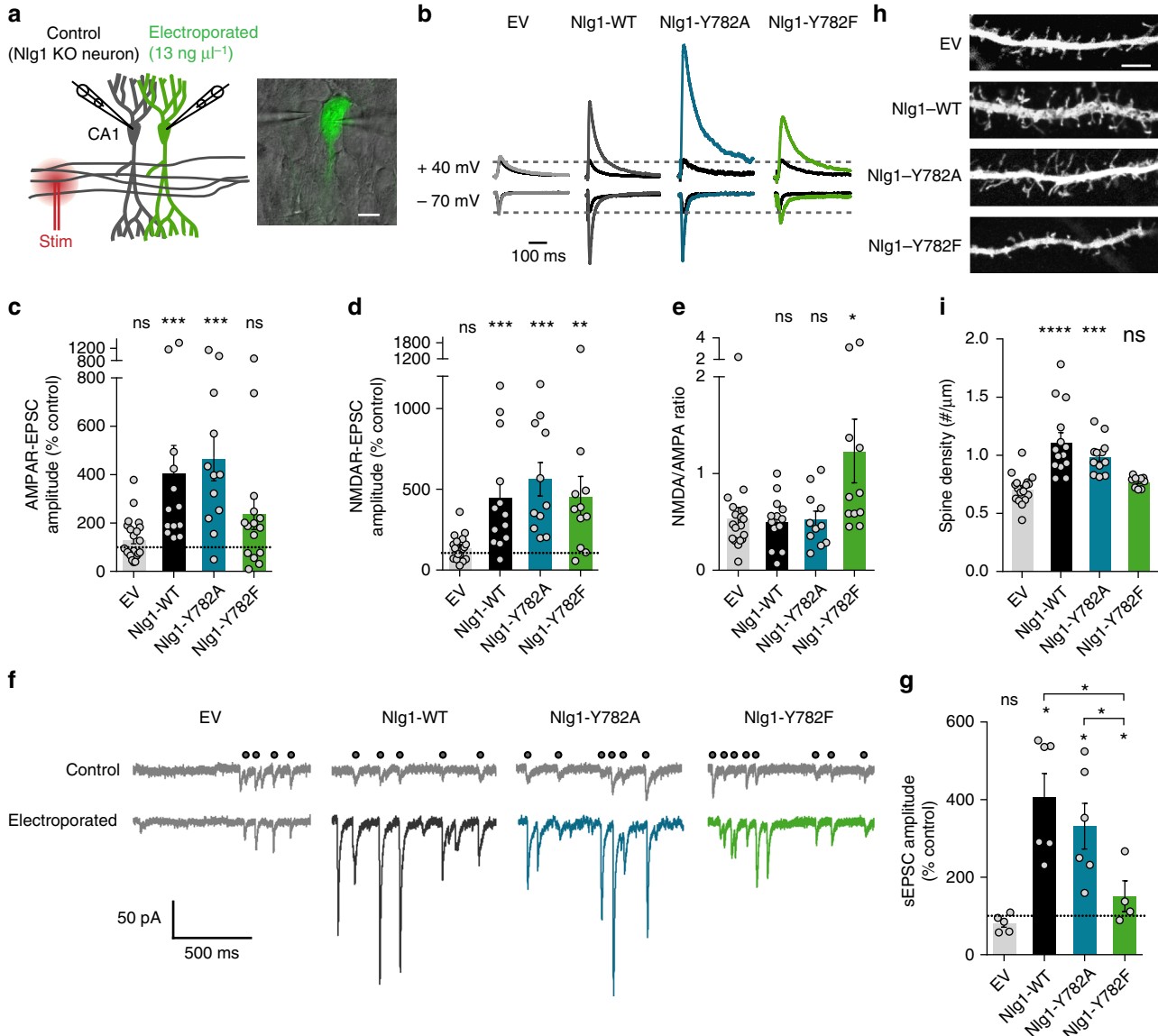

**Fig. 4** Nlg1 mutants differently affect excitatory synaptic transmission and spine density. CA1 neurons of organotypic hippocampal slices from Nlg1 KO mice were single-cell electroporated at DIV 3–5 with GFP plus EV, Nlg1-WT, Nlg1-Y782A, or Nlg1-Y782F, then processed for electrophysiology or immunostaining and confocal imaging at DIV 13–15. **a** Dual whole-cell recording configuration (left) and corresponding image from an experiment (right). Scale bar, 10 μm. **b** Representative traces of evoked AMPAR- and NMDAR-mediated EPSCs recorded at −70 mV and +40 mV, respectively. Color sample traces correspond to electroporated neurons in the different conditions, and black traces correspond to control, unelectroporated neurons. **c**, **d** Average AMPAR- and NMDAR-mediated EPSC amplitudes, respectively, normalized to the control condition (the dashed line indicates 100%, dot plots corresponds to different pairs, from 4 independent experiments). **e** Average ratio between paired NMDAR- and AMPAR-mediated EPSCs. **f** Representative traces of spontaneous EPSCs recorded at −70 mV from electroporated and non-electroporated neurons. Black dots indicate synchronous events. **g** Average of paired spontaneous EPSC amplitudes recorded in electroporated neurons, normalized to the control condition (dashed line at 100%, dot plots corresponds to different pairs, from 3 independent experiments). **h** Representative confocal images of secondary apical dendritic segments from CA1 pyramidal neurons co-expressing GFP along with EV, Nlg1-WT, Nlg1-Y782A, or Nlg1-Y782F. Scale bar, 5 μm. **i** Average spine density for the same conditions (dot plots corresponds to different cells, from 4 independent experiments). Data from graphs **c**, **d**, and **g** were compared to the control condition (non-electroporated neurons) by a Wilcoxon matched-pairs signed rank test. Data in graph **e** were compared by a Kruskal–Wallis test followed by Dunn's multiple comparison test. Data in graphs **g** and **i** were compared by one-way ANOVA followed by Tukey's multiple comparison test (ns: not significant, *$P < 0.05$, **$P < 0.01$, ***$P < 0.001$, ****$P < 0.0001$). Data represent mean ± SEM

**Effects of Nlg1 mutants on dendritic spine density**. We next tested whether the functional effects of the Nlg1 mutants were accompanied by changes in postsynaptic morphology. We thus measured the dendritic spine density of CA1 neurons by confocal microscopy, focusing on the stratum radiatum region (Fig. 4h). The number of spines per unit dendrite length was increased by

~50% in neurons expressing Nlg1-WT compared to cells expressing EV (Fig. 4i), consistent with the reported effects of Nlg1 overexpression in slices[20,33]. While Nlg1-Y782A induced a similar increase of spine density in CA1 cells, this effect was prevented by Nlg1-Y782F (Fig. 4h, i) and by Nlg1 C-terminal truncation mutants (Fig. 5e, f). Since all Nlg1 mutants have the

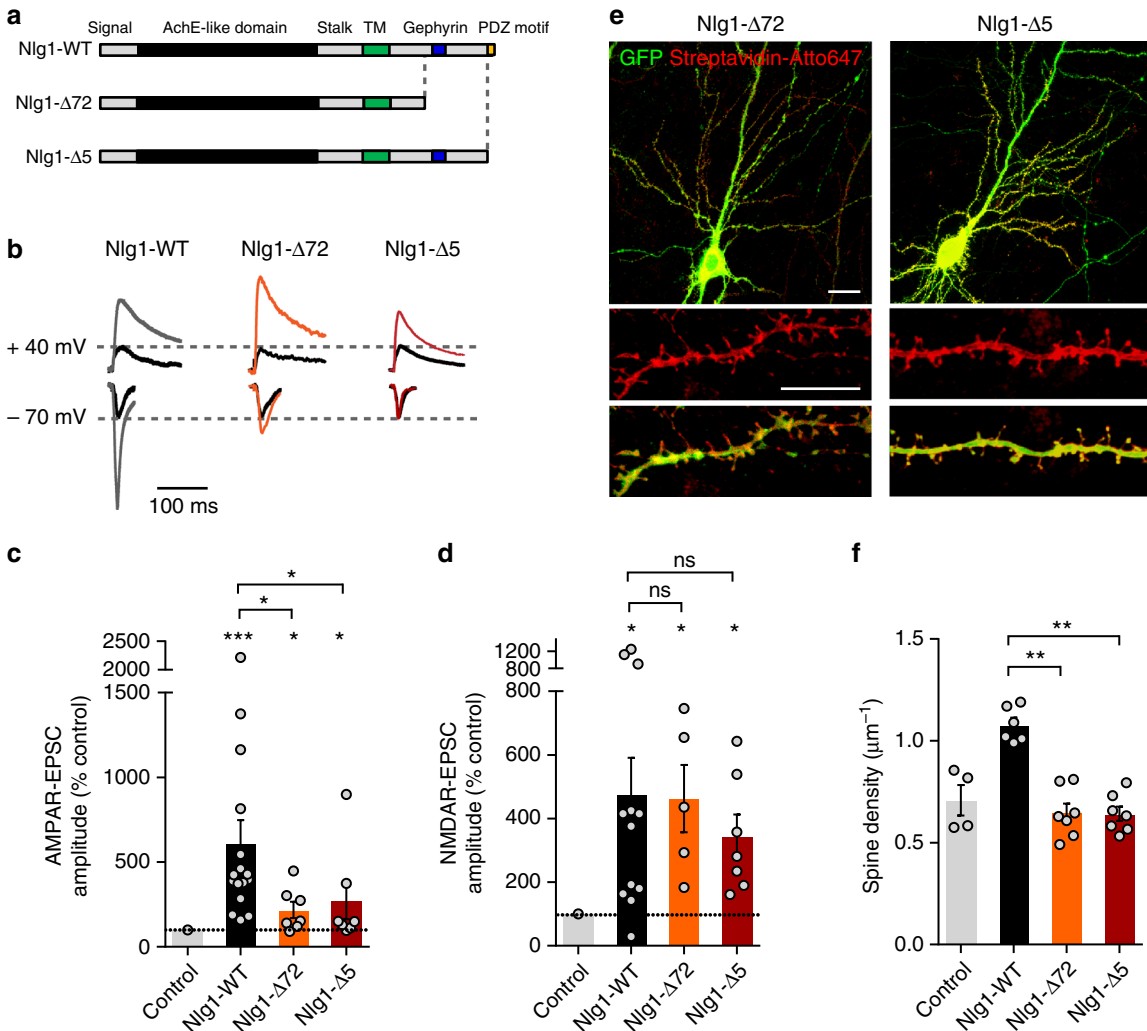

**Fig. 5** Nlg1 C-terminal truncation affects AMPAR-dependent synaptic transmission and spine density. **a** Diagram of Nlg1 truncation mutants lacking the last 5 aa comprising the C-terminal PDZ domain-binding motif (Nlg1-Δ5) or the last 72 amino acids also including the gephyrin-binding motif (Nlg1-Δ72). TM: transmembrane domain. **b** Representative traces of evoked AMPAR- and NMDAR-mediated EPSCs recorded at −70 mV and +40 mV, respectively. Color sample traces correspond to neurons co-electroporated with GFP and Nlg1-WT, Nlg1-Δ5, or Nlg1-Δ72, respectively, while black traces correspond to control, unelectroporated neurons. **c**, **d** Average AMPAR- and NMDAR-mediated EPSC amplitudes, respectively, normalized to control (the dashed line indicates 100%, dot plots corresponds to different pairs, from 3 independent experiments). **e** Confocal images showing CA1 neurons from Nlg1 KO organotypic slices electroporated with GFP (green), BirA$^{ER}$, and AP-tagged Nlg1-Δ5 or Nlg1-Δ72 which were labeled with streptavidin-Atto647 (red). Scale bars, 30 μm (upper panels) and 10 μm (lower panels). **f** Average spine density for CA1 neurons electroporated with EV, Nlg1-WT, Nlg1-Δ5, or Nlg1-Δ72 (dot plots corresponds to different cells, from 2 independent experiments). Data in graphs **c** and **d** were compared to the control condition by Wilcoxon matched-pairs signed rank test, and between themselves using one-way ANOVA followed by Tukey's multiple comparison (ns: not significant, *$P < 0.05$, ***$P < 0.001$). Data in graph **f** were compared by a Kruskal–Wallis test followed by Dunn's multiple comparison test (**$P < 0.01$). Data represent mean ± SEM

same extracellular domain, these results indicate that the morphological differentiation of dendritic spines involves PDZ-mediated intracellular interactions regulated by Y782.

**Both Nlg1-Y782A and Nlg1-Y782F impair long-term potentiation.** Given the strong phenotypes of the Nlg1 point mutants on basal AMPAR-mediated synaptic transmission, we next examined whether they could also affect LTP. Since LTP is reduced in Nlg1 KO mice[14,36], we used organotypic hippocampal slices from Nlg1 WT mice, in which control unelectroporated neurons would be expected to have normal LTP. To reduce the effects of Nlg1 overexpression that could also alter LTP[37], we co-electroporated CA1 neurons with shRNA to Nlg1 containing a GFP reporter + shRNA-resistant AP-tagged Nlg1-WT, -Y782A,

or -Y782F + BirA$^{ER}$ for subsequent labeling with streptavidin (Fig. 6a, b). Neurons expressing the Nlg1-WT along with the shRNA exhibited AMPAR- and NMDAR-EPSC amplitudes in the same range as neighboring neurons expressing endogenous levels of Nlg1 (Fig. 6c–e). In contrast, electroporating the shRNA alone (which knocks down endogenous Nlg1 expression by ~70% in primary hippocampal cultures[28]) decreased by 70% the amplitude of EPSCs relatively to the control neuron, while electroporating recombinant Nlg1 alone (overexpression) induced a 300% increase of EPSC amplitudes. Overall, these results validated our Nlg1 replacement strategy.

In contrast to Nlg1-WT, electroporation of Nlg1-Y782A along with the shRNA induced a 400% increase of AMPAR-EPSCs compared to control neurons, while electroporating Nlg1-Y782F did not increase EPSCs, suggesting that under replacement

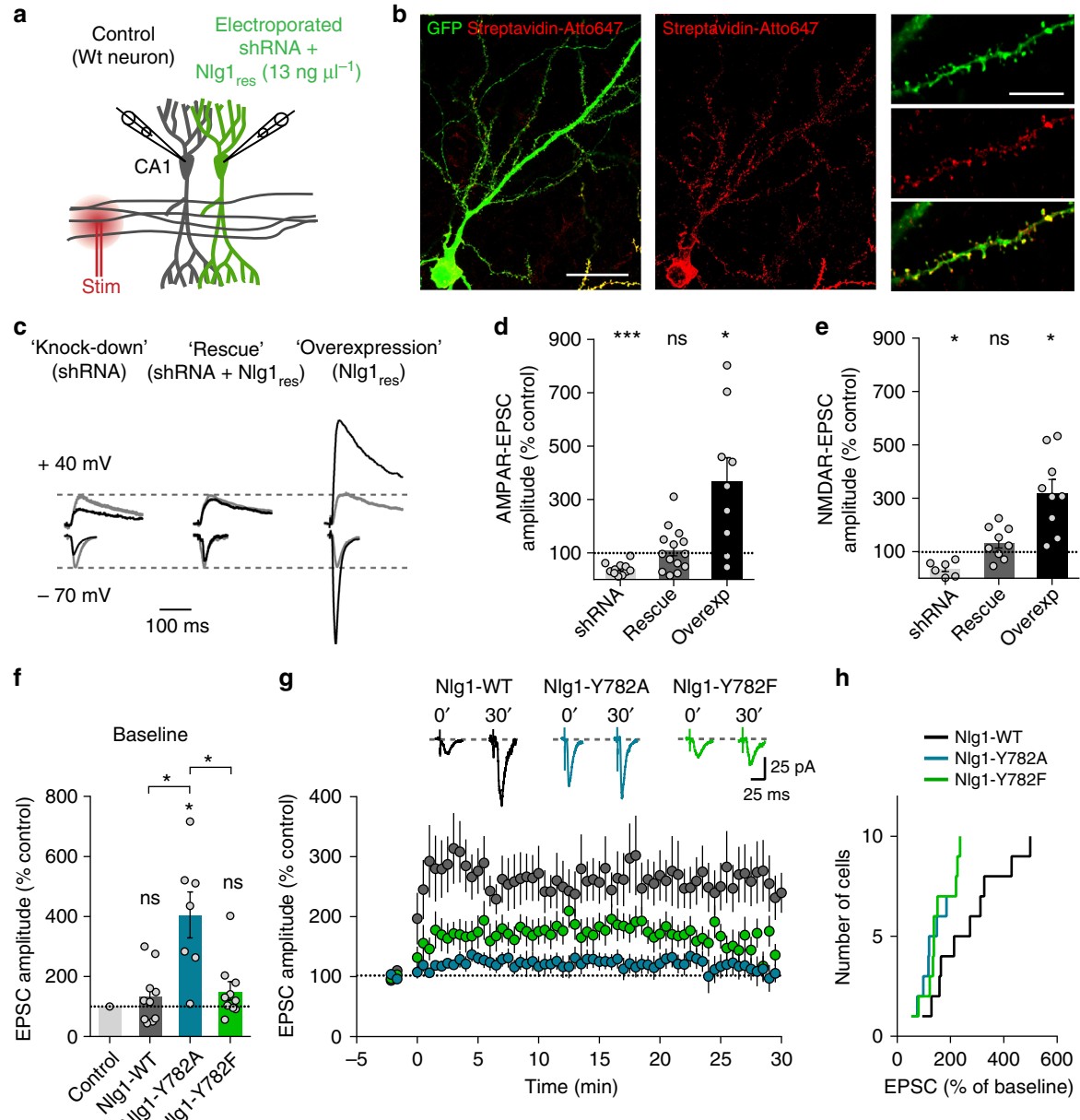

**Fig. 6** Nlg1 replacement by Nlg1 point mutants and LTP experiments. **a–e** Nlg1 replacement strategy. **a** CA1 neurons of organotypic hippocampal slices from wild-type mice were single-cell electroporated at DIV 3–5 with either Nlg1 shRNA alone (knockdown), Nlg1 shRNA + resistant Nlg1-WT (rescue), or with resistant Nlg1-WT alone (overexpression). **b** Representative confocal images of a CA1 neuron co-expressing the GFP reporter from the shRNA (green) and biotinylated AP-Nlg1 stained with streptavidin-Atto647 (red). Scale bars, 40 μm (left panels) and 10 μm (right panels). **c** Representative traces of evoked AMPAR and NMDAR-mediated EPSCs recorded at −70 mV and +40 mV, respectively, in electroporated (black) and control (gray) neurons. **d**, **e** Average AMPAR- and NMDAR-mediated EPSCs, respectively, in the three conditions (dot plots corresponds to different pairs, from 3 independent experiments). **f** Average basal AMPAR-mediated EPSCs in CA1 cells co-expressing shRNA to Nlg1 plus Nlg1-WT, -Y782A, or -Y782F rescue constructs (dot plots corresponds to different pairs, from 5 independent experiments). **g** Average AMPAR-mediated EPSCs for the three conditions, upon LTP induction at time 0. Sample traces are shown at time 0 and 30 min after LTP induction. **h** Cumulative distribution of the long-term plateau of AMPAR-mediated EPSC in the three conditions, expressed as a percentage of the baseline level. Data in graphs **d–g** represent mean ± SEM and were compared to the control condition (unelectroporated) by Wilcoxon matched-pairs signed rank test (*$P < 0.05$, ***$P < 0.001$, ns: not significant compared to control)

conditions, these two Nlg1 mutants differently retain AMPARs (Fig. 6f). Using a pairing protocol to elicit LTP, neurons expressing Nlg1-WT showed a ~250% increase in AMPAR-mediated EPSCs during at least 30 min, similarly to control unelectroporated neurons (Supplementary Fig. 5), again validating the replacement paradigm. In contrast, neurons expressing Nlg1-Y782F displayed a significantly reduced LTP, in accordance

with their difficulty to recruit PSD scaffolds and AMPARs (Fig. 6g, h). Strikingly, neurons expressing Nlg1-Y782A did not show any LTP, consistent with the capacity of this Nlg1 mutant to recruit AMPARs in a constitutive manner, thereby blocking further potentiation (Fig. 6g, h). Thus, Nlg1 Y782 is not only important for AMPAR-dependent basal synaptic transmission, but also for synaptic plasticity.

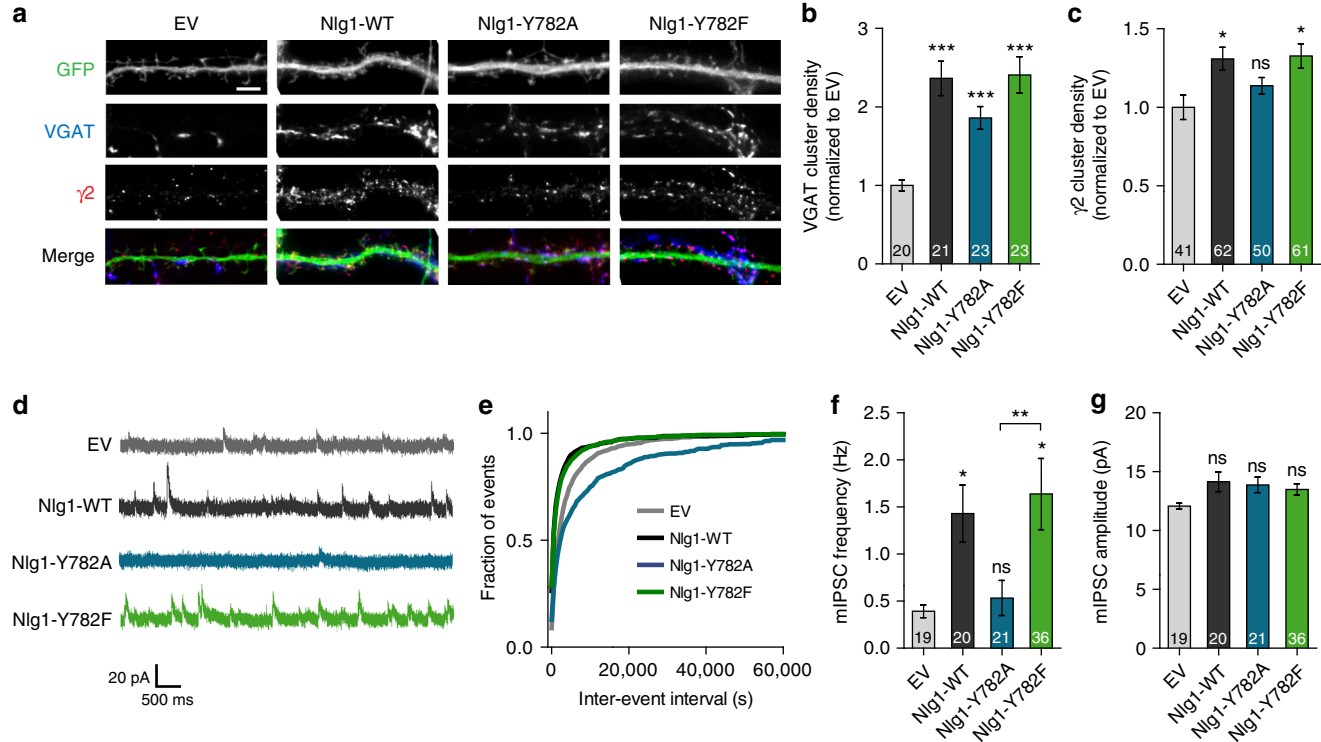

**Fig. 7** Expressing Nlg1-Y782F but not Nlg1-Y782A increases the number of functional inhibitory synapses. **a** Dendrites from DIV 14–15 dissociated neurons transfected with EV, Nlg1-WT, -Y782A, or -Y782F along with GFP (green). Neurons were live stained with an antibody against the γ2 subunit of GABA$_A$ receptors (red), then fixed and counterstained with an antibody against VGAT (blue). Scale bar, 5 μm. **b**, **c** Density of VGAT- or γ2-positive clusters, respectively, normalized to the EV condition (number of cells indicated within the bars, from 3 independent experiments). **d** Representative traces of mIPSCs recordings from DIV 14–15 neurons expressing EV, Nlg1-WT, -Y782A, or -Y782F clamped at +10 mV in presence of TTX and NBQX. **e** Cumulative distributions of the mIPSCs inter-event intervals for the corresponding conditions. **f**, **g** Mean mIPSC frequencies and amplitudes, respectively, for each condition (number of cells indicated within the bars, from 4 independent experiments). Data in graphs **b**, **c**, **f**, and **g** represent mean ± SEM and were compared by a Kruskal–Wallis test followed by Dunn's multiple comparison test (ns: not significant, *P < 0.05, **P < 0.01, ***P < 0.001)

**Effect of Nlg1 mutants on inhibitory synapse differentiation.** Since Nlg1 binds gephyrin in vitro (Fig. 1c) and can occasionally be found at inhibitory synapses[38], we next investigated the effect of Nlg1-Y782 mutants on GABAergic differentiation. We first examined the distribution of VGAT and GABA$_A$ receptors by immunocytochemistry in primary hippocampal neurons. Nlg1-WT expression increased by 2-fold the density of VGAT clusters and by ~25% the density of GABA$_A$ clusters, compared to control neurons expressing EV (Fig. 7a–c). While both Nlg1 mutants recruited VGAT clusters similarly to Nlg1-WT, Nlg1-Y782F but not Nlg1-Y782A, significantly promoted the increase in GABA$_A$ cluster density (Fig. 7a–c). We also performed single molecule tracking of GABA$_A$ receptors at the cell surface of cultured hippocampal neurons using anti-γ2 antibodies bound to Atto-594-conjugated anti-rabbit Fab. Neurons expressing Nlg1-WT or Nlg1-Y782A exhibited slightly reduced GABA$_A$ receptor diffusion compared to EV, while neurons expressing Nlg1-Y782F showed a significant reduction (Supplementary Fig. 6a-c), consistent with the stronger recruitment of gephyrin and GABA$_A$ receptors by this mutant (Fig. 1e, f and Supplementary Fig. 6d).

At the functional level, we examined the effects of Nlg1 mutants on GABAergic transmission by recording miniature inhibitory postsynaptic currents (mIPSCs) in primary hippocampal cultures. Compared to EV, Nlg1-WT expression tripled the frequency of mIPSCs[10], but did not affect their amplitude (Fig. 7d–g), consistent with an increase in GABAergic inputs on transfected cells (Fig. 7a–c). Strikingly, Nlg1-Y782F increased the mIPSC frequency similarly to Nlg1-WT, whereas Nlg1-Y782A failed to induce this effect (Fig. 7d–f). These results reveal that

Nlg1-Y782F, but not Nlg1-Y782A, favors the assembly of functional inhibitory synapses in primary neurons. When the ratio between mEPSC and mIPSC frequency was computed on a per cell basis, Nlg1-WT lay in between Nlg1-Y782A and –Y782F (Supplementary Fig. 7a), suggesting that Nlg1-WT harbors an intermediary tyrosine phosphorylation level.

We next aimed to confirm these differential effects of Nlg1 mutants on inhibitory synaptic transmission in organotypic slices. We expressed Nlg1-WT, -Y782A, or -Y782F in CA1 neurons from Nlg1 KO slices, and recorded GABA$_A$ receptor mediated currents upon stimulation of inhibitory inputs[39]. Surprisingly, all Nlg1 constructs induced a similar ~5-fold enhancement of IPSCs compared to non-electroporated neighboring neurons, independently of the Y782 mutation (Supplementary Fig. 7b, c). These results indicate that Nlg1 tyrosine point mutants act differently on inhibitory synapse differentiation in dissociated cultures and organotypic slices, potentially because of variations in the localization and molecular composition of inhibitory inputs between the two systems[40].

**Screen of the tyrosine kinases phosphorylating Nlg1.** We then sought to characterize the mechanism of Nlg1 tyrosine phosphorylation and manipulate it, independently of expressing Nlg1 mutants. To assess whether Nlg1 phosphorylation was regulated during development, we measured the phosphotyrosine (pTyr) levels of immunoprecipitated Nlg1 after increasing culture periods (DIV 7–8, 10–11, 14, and 16–17). The pTyr signal showed an increase around DIV 10 (Supplementary Figure 8a, b), which

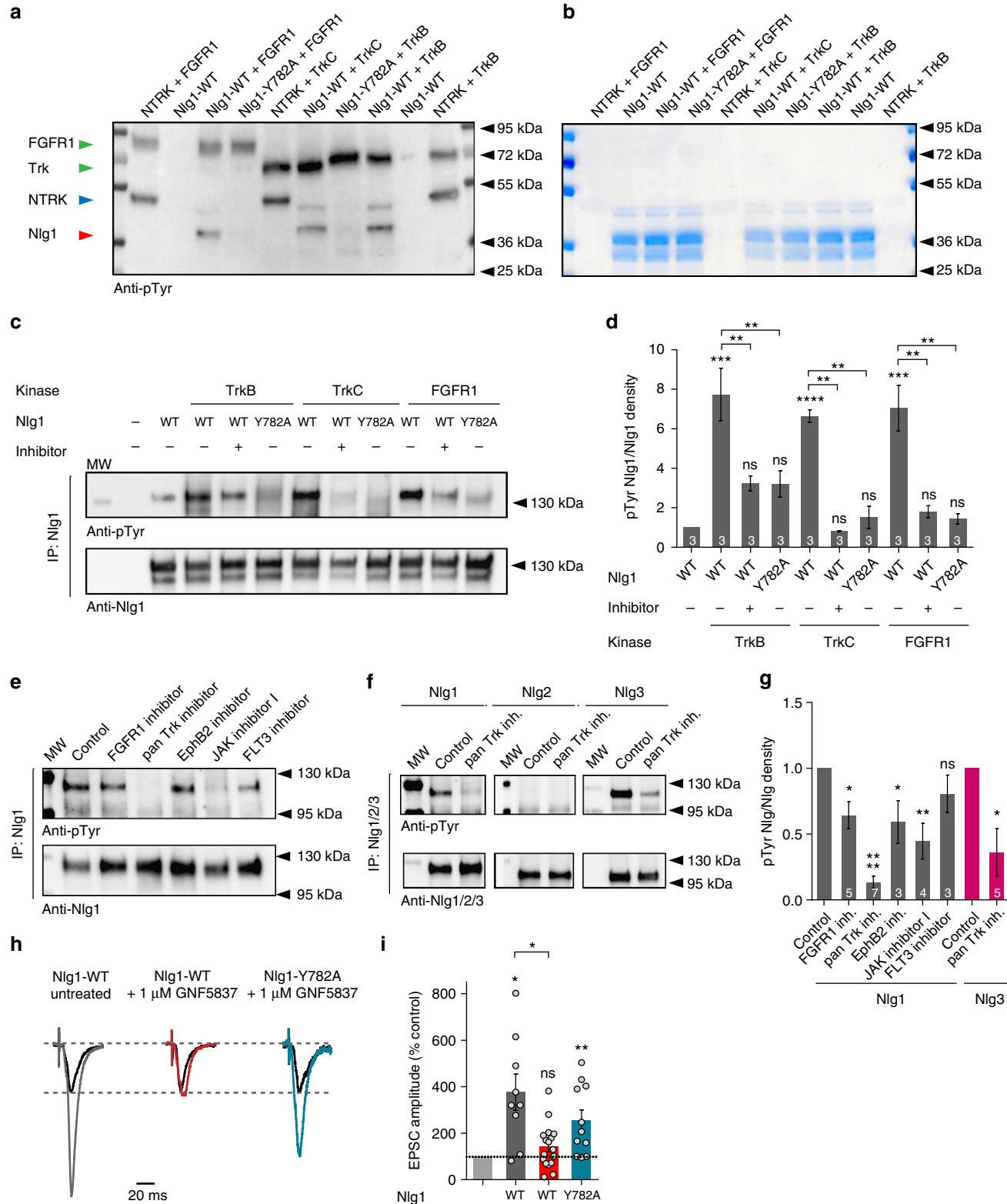

coincides with a peak of synaptogenesis[41]. We then searched for kinases that phosphorylate Nlg1. An initial commercial screen of purified tyrosine kinases able to phosphorylate 16-aa Nlg1 peptides encompassing the gephyrin-binding motif led to several candidates, including the FGF receptor 1 (FGFR1), JAK-2, TrkB, TrkC, and EphB2 (Supplementary Fig. 8c). Using an in vitro kinase assay with recombinant GST proteins fused to the

intracellular domain of Nlg1, followed by pTyr immunoblots, we confirmed that FGFR1, TrkB, and TrkC (Fig. 8a, b), but not JAK-2 (Supplementary Fig. 8d), directly phosphorylated Nlg1-WT. Importantly, GST-Nlg1-Y782A was not phosphorylated in this assay, demonstrating that Y782 is the unique phosphorylation site in the Nlg1 intracellular domain. We also checked by biochemistry that recombinant FGFR1, TrkB, and TrkC all

**Fig. 8** Identification of tyrosine kinases that phosphorylate Nlg1. **a**, **b** In vitro kinase assay. Purified tyrosine kinases (GST-FGFR1, -TrkB, or -TrkC) were incubated in the presence of ATP with GST-Nlg1, GST-Nlg1-Y782A, or a positive substrate (NTRK), and run on a polyacrylamide gel. **a** pTyr immunoblot. Note the phosphorylation of NTRK (blue arrow) and GST-Nlg1 (red arrow) but not GST-Nlg1-Y782A by the three kinases. Note the autophosphorylation of the three kinases (green arrow). **b** Corresponding Coomassie gel. GST-Nlg1 shows several bands, corresponding to partial degradation. The strongest band at 40 kD contains the gephyrin-binding motif. The NTRK and kinases are present in low amounts and barely appear on the gel. **c** Protein extracts from COS cells expressing Nlg1-WT, Nlg1-Y782A, with or without FGFR1, TrkB, or TrkC, were immunoprecipitated with Nlg1 antibodies. Some cells were pretreated with FGFR1 or pan-Trk inhibitors. pTyr and Nlg1 immunoblots are shown. **d** pTyr signals normalized to Nlg1 levels in the different conditions (number of experiments within bars). **e**, **f** Protein extracts from cortical cultures at DIV 10–14 pretreated with the kinase inhibitors for 24 h were immunoprecipitated with Nlg1, Nlg2, or Nlg3 antibodies. pTyr and Nlg1/2/3 immunoblots are shown, respectively. **g** Average pTyr signals for the different inhibitors, normalized to the control, untreated condition (number of experiments within bars). **h** CA1 neurons from Nlg1 KO slices were electroporated with Nlg1-WT or Nlg1-Y782A, and AMPAR-mediated EPSCs were recorded upon stimulation of Schaffer's collaterals, in comparison to unelectroporated neighboring neurons (black traces). Organotypic cultures were treated or not with 1 μM GNF5837 for 7 days before the recordings. **i** Average AMPAR-mediated EPSCs amplitude in the 3 conditions, normalized to non-electroporated controls (number of pairs indicated within the bars, from 3 independent experiments). Data in graphs **d** and **g** were compared by one-way ANOVA followed by Bonferroni post hoc test (ns: not significant, $*P < 0.05$, $**P < 0.01$, $****P < 0.001$). Data in graph **i** were compared to the control condition by Wilcoxon matched-pairs signed rank test, and between themselves using one-way ANOVA followed by Tukey's multiple comparison (ns: not significant, $*P < 0.05$, $**P < 0.01$). Data represent mean ± SEM

phosphorylated HA-Nlg1 in COS-7 cells, and that the corresponding kinase inhibitors (PD166866 and GNF5837, respectively) blocked Nlg1 phosphorylation (Fig. 8c, d). To determine which kinase(s) phosphorylate(s) endogenous Nlg1, we used those and other kinase inhibitors in primary neurons followed by Nlg1 immunoprecipitation and pTyr immunoblot. When applied for 24 h, inhibitors of FGFR1, EphB2, JAK, or FLT3 reduced by varying degrees (25–55%) the Nlg1 pTyr level, while the pan-Trk inhibitor GNF5837 almost completely abolished Nlg1 phosphorylation (Fig. 8e–g), suggesting that Trk family members are major regulators of Nlg1 phosphorylation in neurons.

To examine whether Nlg1 phosphorylation by Trk family members could effectively modulate the effect of Nlg1-WT expression on synaptic transmission, we incubated hippocampal slices from Nlg1 KO slices with those inhibitors, and measured AMPAR-mediated EPSCs in CA1 neurons expressing Nlg1-WT or Nlg1-Y782A (Fig. 8h, i). Strikingly, the strong increase in EPSCs amplitude induced by Nlg1-WT expression (~400% of the non-electroporated control) was completely abolished when incubating the slices with the pan-Trk inhibitor, but unaffected by the FGFR1 inhibitor (Fig. 8h, i and Supplementary Fig. 9). Importantly, neurons expressing Nlg1-Y782A were resistant to GNF5837, demonstrating that the increase of synaptic responses caused by Nlg1 expression involves Y782 phosphorylation by Trks.

**The tyrosine phosphorylation mechanism is specific to Nlg1.** Finally, given that the gephyrin-binding motif containing the critical tyrosine is highly conserved among all Nlg isoforms[23], we examined whether Nlg2 and Nlg3 could also be phosphorylated (the corresponding positions are Y770 and Y792, respectively). In COS cells, both Nlg2 and Nlg3 where phosphorylated by recombinant tyrosine kinases such as TrkB/C (Supplementary Fig. 8e). However, only Nlg1 and Nlg3 -but not Nlg2- were tyrosine phosphorylated in neurons (Fig. 8f), likely reflecting the preferential presence of endogenous tyrosine kinases such as Trks at excitatory synapses[38,42]. Compared to Nlg1, the Nlg3 pTyr signal in neurons was only partially blocked by the pan-Trk inhibitor, suggesting that an additional tyrosine specific to Nlg3 (Y827) can also be phosphorylated. Indeed, a Nlg3 construct bearing a Y792A mutation was still significantly phosphorylated in COS-7 cells compared to Nlg3-WT (Supplementary Fig. 10a-c). Since Nlg3 is also present at glutamatergic synapses[17], we investigated whether mutations of the conserved tyrosine in Nlg3 (Y792) affected miniature AMPA currents (Supplementary Fig. 10d). In contrast to Nlg1, the two Nlg3 phosphomutants (Y792F and Y792A) equally increased the frequency of AMPAR-

mediated mEPSCs compared to neurons expressing EV (Supplementary Fig. 10d-f). Taken together, these results indicate that the regulation of glutamatergic synapse differentiation by Nlg tyrosine phosphorylation is specific to the Nlg1 isoform.

## Discussion

We demonstrate that a critical tyrosine located in the intracellular tail of Nlg1 regulates the differentiation of functional excitatory synapses. We propose a model in which Nlg1 phosphorylation through Trk tyrosine kinases at newly forming synapses, prevents gephyrin binding and instead promotes the C-terminal recruitment of PSD scaffolding proteins, which serve as anchors for the recruitment of AMPARs in basal conditions and during potentiation (Fig. 9).

Immunofluorescence data and electrophysiological recordings in dissociated neurons and organotypic slice cultures show that Nlg1 expression is able to induce both excitatory and inhibitory synapse differentiation, as previously reported[10,12,30,43]. However, the two Nlg1 tyrosine point mutants showed a strong discrimination in their ability to assemble synapses: Nlg1-Y782A essentially formed excitatory synapses endowed with postsynaptic PSD-95 and AMPARs assembled in front of VGlut1 positive terminals. In contrast, Nlg1-Y782F was unable to recruit AMPARs in front of VGlut1 positive terminals, suggesting that Nlg1 phosphorylation might be the mechanism by which Nlg1 recruits AMPARs[21,41]. Instead, Nlg1-Y782F primarily formed inhibitory synapses containing gephyrin and GABA_A receptors, in front of VGAT positive pre-synapses in dissociated cultures. In a way, the Nlg1-Y782F mutant resembles the Nlg2 isoform primarily located at inhibitory synapses[15], which harbors a specific interaction motif with the membrane-interacting collybistin, reinforcing Nlg2 binding to gephyrin[22,23]. However, a recent study showed that the deletion of intracellular collybistin- or gephyrin-binding motifs does not prevent the increase of inhibitory synaptic transmission induced by Nlg2 expression, and therefore rather supports a role for the Nlg2 extracellular domain[39]. Our recordings of inhibitory currents in organotypic slices from Nlg1 KO mice failed to reveal a difference between Nlg1-Y782A and -Y782F mutants, potentially because those mutants can form heterodimers with endogenous Nlg2 or Nlg3[44], or because inhibitory inputs targeting primarily the CA1 cell soma and proximal dendrites are structurally and functionally different than inhibitory synapses formed in dissociated cultures.

Between these extreme Nlg1 and Nlg2 isoforms, Nlg3 is associated with both excitatory and inhibitory synapses[17]. Surprisingly, it was suggested that the increase in AMPAR- and NMDAR-mediated synaptic transmission induced by Nlg3

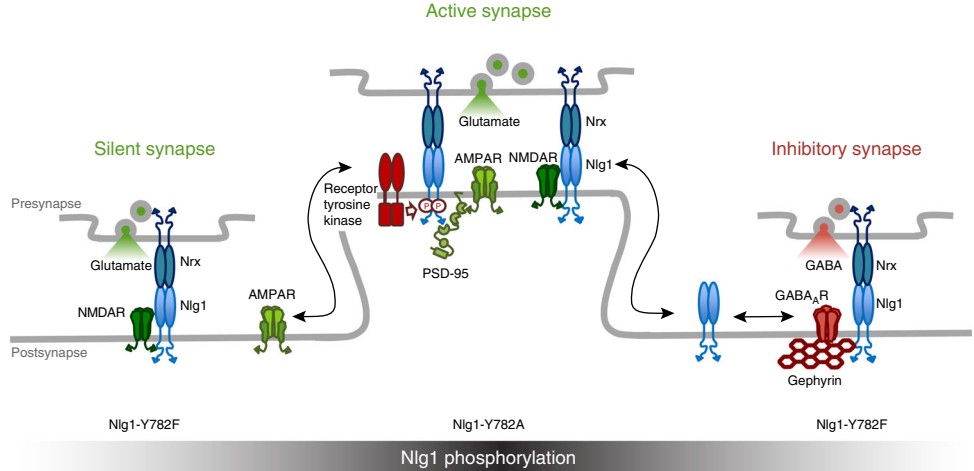

**Fig. 9** Working model for the role of Nlg1 tyrosine phosphorylation in synapse differentiation and potentiation. Phosphorylated Nlg1 (mimicked by Nlg1-Y782A) preferentially recruits PSD-95 and AMPARs and favors dendritic spines. In contrast, non-phosphorylated Nlg1 (mimicked by Nlg1-Y782F) diffuses out of dendritic spines and interacts preferentially with gephyrin, thus preventing PSD-95 and AMPAR recruitment and resulting in silent synapses. NMDARs are recruited independently of mutations in the Nlg1 intracellular domain, consistent with a direct extracellular coupling to Nlg1. A fraction of Nlg1-Y782F associates with gephyrin scaffolds and GABA$_A$ receptors to make inhibitory synapses

overexpression in CA1 cells was neither dependent on the C-terminal PDZ domain- nor the gephyrin-binding motifs, but on a non-canonical region closer to the transmembrane domain (aa -77–90), whose binding partner has not been identified yet[20]. This finding seems at odds with our results showing that Nlg1-Y782F, Nlg1-Δ5, and Nlg1-Δ72 all exhibited strongly reduced AMPAR-mediated synaptic responses compared to Nlg1-WT, but similar NMDAR-mediated synaptic transmission. One explanation for this discrepancy might lie in the Nlg replacement strategies, i.e., triple Nlg1-3 knockdown[20] vs. single Nlg1 KO (this study), where the competition between neurons for synapse formation based on relative Nlg levels, might be different[33]. This selective effect of the Nlg1 C-terminal truncation on AMPAR-mediated currents is likely to depend on the recruitment of PSD-95, which is prevented by these Nlg1 mutants[12,19,21,45]. Interestingly, a loss of PSD-95, and not of NMDARs was observed upon chronic treatment of neurons with CRIPT PDZ binding ligands competing for the interaction between Nlg1 and the PDZ domain 3 of PSD-95[24,46]. The residual increase in AMPAR-mediated responses observed with the three Nlg1 mutants agrees with the formation of heterodimers between mutated Nlg1 and endogenous Nlg3[9,20,44]. The mechanisms by which Nlg1 and Nlg3 regulate the differentiation of glutamatergic synapses seem different since, although endogenous Nlg3 is highly phosphorylated, the Nlg3-Y792F mutant is still able to strongly increase the frequency of AMPAR-mediated mEPSCs. Together with the absence of Nlg2 pTyr signal in neurons, those data indicate that the tyrosine phosphorylation regulatory mechanism is specific to the Nlg1 isoform.

The two Nlg1 point mutants bear the same extracellular domain, and should thus be equally competent to recruit excitatory and inhibitory inputs. However, there was a trend for the density of VGlut1 puncta to be higher on neurons expressing Nlg1-Y782A than Nlg1-Y782F, whereas the density of VGAT puncta was higher on neurons expressing Nlg1-Y782F than Nlg1-Y782A. This observation suggests that the intracellular mutations may have retrograde effects on presynaptic differentiation[35]. Alternatively, an activity-dependent validation mechanism is possible, by which presynaptic terminals that match the respective postsynaptic receptors are retained, while those that do not match (i.e., silent and/or mute synapses) tend to be eliminated. In

this context, NMDAR activity is required for the maintenance of excitatory synapses on Nlg1-overexpressing cells[26]. This mechanism might also regulate Nlg1 trafficking and accumulation at glutamatergic synapses through the phosphorylation of T738 by CamKII[34]. Importantly, LTP was reduced in neurons expressing Nlg1-Y782F, and occluded in neurons expressing Nlg1-Y782A. These results support the concept that Nlg1 tyrosine phosphorylation controls synapse unsilencing in an activity-dependent manner. This mechanism may involve the trapping of extrasynaptic diffusing AMPARs induced by an increase in binding affinity between AMPAR auxiliary subunits and the PSD scaffold[32,47,48]. The fact that the two Nlg1 mutants impaired LTP although they increased NMDAR-mediated EPSCs similarly to Nlg1-WT, is consistent with the observation that conditional ablation of Nlg1 in CA1 neurons alters LTP in an NMDAR-independent pathway[14]. Another interesting feature is that synapses generated by the Nlg1-Y782F mutant can recruit both GABA$_A$ receptors (through gephyrin) and NMDA receptors, most likely on the dendritic shaft. While our data indicate that these receptors are assembled in front of GABAergic vs. glutamatergic release sites, respectively, they could also coexist within the same postsynaptic complex. Interestingly, a fraction of synaptic NMDARs (15–20%) can be found in GABAergic synapses during development, where they may potentiate the depolarizing function of GABAergic synapses[49], and in the mature hippocampus where they trigger retrograde nitric oxide signaling[50].

Apart from scaffold and receptor recruitment, Nlg1 also regulated dendritic spine morphology. The ability to induce new spines through local glutamate uncaging was shown to depend on the relative Nlg1 expression level[33], and spines that do not recruit PSD-95 quickly disappear in vivo[51]. Here, Nlg1-Y782F as well as Nlg1-Δ5 and -Δ72 mutants, were all unable to increase spine density, supporting the concept that molecular interactions involving the Nlg1 C-terminus are crucial to form or maintain new spines. Furthermore, the inability of those Nlg1 mutants to recruit AMPARs might contribute to this process since the N-terminal domain of GluA2 regulates spine density[52]. In contrast, Nlg1-Y782A, which interacts very well with PSD-95 and recruit AMPARs, was competent for spine formation. Alternatively, the effects of Nlg1 on spine density may originate from the

stabilization of dendritic filopodia, through a connection of Nlg1 to the branched actin network[53,54]. Indeed, Nlg1 has an intracellular binding motif to the WAVE complex[55], and WAVE is localized at the PSD where it nucleates new branched actin filaments through Arp2/3 activation[56]. Additionally, NMDAR-dependent activity can induce Nlg1 proteolytic cleavage[57,58] and, in turn, spinogenesis is stimulated by the presence of the cleaved Nlg1 intracellular fragment which binds to SPAR, promotes cofilin phosphorylation and actin stabilization[59].

The data obtained with Nlg1 mutants provide converging evidence that Nlg1 Y782 phosphorylation is required for glutamatergic synapse development. Although the Y782A/F mutations are not traditionally phospho-mimetic, they strongly impacted the binding of Nlg1 to gephyrin both in vitro and in neurons. The Y782A mutant phenocopies phosphorylated Nlg1, since the addition of a phosphate group as well as the alanine mutation block gephyrin binding, while Nlg1-Y782F mimics unphosphorylated tyrosine and strongly binds gephyrin[25]. Our view is that binding of Nrx1β to Nlg1 at early glutamatergic contacts triggers tyrosine phosphorylation, which makes Nlg1 unable to bind gephyrin and instead promotes binding to PSD-95[25]. The higher pTyr level of Nlg1 around DIV 10, precisely when initial axon-dendrite contacts mature into functional synapses[41], supports this concept. It is likely that other Nrx isoforms (i.e., α-Nrxs) or splice variants (e.g., SS4 + ) present in GABAergic axons bind less well to Nlg1 and thus do not induce tyrosine phosphorylation, promoting instead gephyrin recruitment and inhibitory post-synapse assembly[29,60,61]. While the Nlg1 intracellular domain is unstructured and gephyrin and PSD-95 binding sites are remote, it is possible that the Nlg1 tail folds around the multimeric gephyrin scaffold, thereby preventing accessibility of its C-terminus to PSD-95 vertically oriented to the plasma membrane[62]. This model predicts that at excitatory synapses, Nrx-bound Nlg1 (best mimicked by Nlg1-Y782A) is in a phosphorylated state and strongly anchored—explaining the similar phenotypes between Nlg1-WT and Nlg1-Y782A—while outside excitatory synapses, Nrx-free Nlg1 (best mimicked by Nlg1-Y782F) would be in a non-phosphorylated state and more diffusive. An interesting underlying mechanism could involve a tripartite complex comprising Nrx, Nlg1, and a receptor tyrosine kinase (RTK), phosphorylating Nlg1 by a combination of proximity and ligand-activation. Indeed, we identified several RTKs able to phosphorylate Nlg1 in vitro and in heterologous cells, including the FGFR1[63], TrkB, TrkC[64], and EphB2[65]. The use of pharmacological kinase inhibitors further demonstrate that Trk family members maintain a substantial level of Nlg1 tyrosine phosphorylation in neurons, and promote the action of Nlg1 on glutamatergic synaptic transmission. To conclude, the Nlg tyrosine phosphorylation mechanism demonstrated here plays an important role in the control of the synapse differentiation and plasticity, with potentially important consequences for cognitive functions.

## Methods

**DNA constructs.** Plasmids for mouse HA-Nlg1 harboring both A and B splice inserts, short hairpin RNA to murine Nlg1 (shNlg1), and mouse Nrx1β-Fc without splice site 4 (SS4-) were gifts from P. Scheiffele (Biozentrum, Basel). The HA-tagged Nlg1-Y782A and -Y782F plasmids made from HA-Nlg1 were described previously[25]. GST-Nlg1-WT, -Y782A, and -Y782F plasmids were generated by amplifying the cytoplasmic tail sequence of HA-Nlg1, HA-Nlg1-Y782A, and HA-Nlg1-Y782F, respectively, with the following primers: 5′CTCGAGAGA-CATGATGTCCACCGG3′ and 5′GCGGCCGGCTCTCTCATCTGGCTATACCC3′ and inserted in the pGex-4T1 vector (Amersham) at the XhoI/NotI sites. Mouse AP-tagged Nlg1 and biotin ligase (BirA^ER) were gifts from A. Ting (Stanford University, CA). AP-Nlg1-Y782A and -Y782F were generated by amplifying the end of the C-terminal sequence of HA-tagged Nlg1-Y782A and -Y782F, respectively, with the following primers: 5′GGGGTACCTCATCTGCATAATC3′ and 5′GTCGACTGCAGAATTCGAAGCTTCTATACCCTGGTTGTTGAATG 3′, and

inserted in the AP-tagged Nlg1 construct at the KpnI/SalI sites. AP-tagged Nlg1-Δ5 was obtained performing a PCR of the end of AP-Nlg1 lacking the last 15 nucleotides with the following primers: 5′GGGGTACCTCATCTGCATAATC3′ and 5′GTCGACCTAATGTGAATGGGGGTGTGGG 3′. The deleted sequence was inserted in AP-Nlg1 at KpnI/SalI sites. AP-tagged Nlg1-Δ72 was generated by deleting the 216 last nucleotides with the In-Fusion® HD Cloning Kit (Takara Bio) with the following primers: 5′CACATGAGTAGAAGCTTCGAATTC3′ and 5′ GCTTCTACTCATGTGGATGGATG3′. shRNA-resistant AP-tagged Nlg1 was described previously[28]. Rescue AP-tagged Nlg1-Y782A and –Y782F were produced by subcloning the C-terminal parts of AP-Nlg1-Y782A and -Y782F, respectively, into rescue AP-tagged Nlg1-WT at KpnI/SalI sites. Mouse HA-Nlg2 with splice insert A was purchased from Addgene. Rat HA-Nlg3 was a generous gift from N. Brose (Max Planck Institute, Göttingen). HA-Nlg3 point mutants Y792A and Y792F were generated from the HA-Nlg3 construct, by site-directed mutagenesis using the QuickChange II XL kit (Agilent Technologies). Homer1c-GFP and Gephyrin-Venus were gifts from S. Okabe (Tokyo University, Japan), and A. Triller (ENS, Paris), respectively. The SEP-GluA1 construct was previously described[21]. FGFR1-Flag was a gift from L. Duchesne (Université de Rennes, France). Rat TrkB-RFP and HA-TrkC were gifts from F. Janneteau (Montpellier, France) and A.M. Craig (Vancouver, Canada), respectively. The pEGFP-C1 vector (Clontech) was used for electroporation or transfection to express soluble GFP as a volume marker, and empty vector (EV) corresponds to pcDNA3.1 hygro(−) (Invitrogen).

**Purified recombinant proteins.** E. Coli BL21 (DE3)-RIPL cells were transformed with each of the GST fusion protein expression plasmids (GST, GST Nlg1-WT, Y782A, and Y782F). Cells were amplified in 0.5 L of LB medium at 37 °C for 2–3 h until the absorbance at 600 nm reached 0.6–0.7. The induction of protein expression was started by adding IPTG (Isopropyl β-D-1-thiogalactopyranoside) at 0.25 mM to the culture and incubating it overnight at 16 °C. Cells were harvested by centrifugation and the pellet was resuspended in 35 mL of lysis buffer (100 mM EDTA, 10% glycerol, 1% TritonX-100, 1 mg/mL lysozyme in PBS, pH 7.4) containing protease inhibitor cocktail III (Calbiochem) and then incubated on a rotating wheel for 30 min at 4 °C. Cells were lysed by ultrasonication (Branson Sonifier 450 at 50% power with a 40% duty cycle for 4 min at 4 °C). After addition of DTT at a final concentration of 10 mM, the lysate was cleared by centrifugation (15,000 × g, 45 min, 4 °C) and filtered at 0.2 μm. GST-fusion proteins were purified with 3 mL of Glutathione Sepharose™ 4 Fast Flow (GE Healthcare) by batch-binding. Bound proteins were washed with 10 volumes of PBS (pH 7.4) and eluted with 20 mL of a 10 mM glutathione in 50 mM Tris (pH 8) solution. Fractions were analyzed by SDS-PAGE (4–15% Mini-PROTEAN TGX Precast Protein Gels, Bio-Rad) followed by Coomassie staining. Fractions containing the proteins were pooled and dialyzed against PBS (pH 7.4). Protein concentrations were measured by determining light absorption at 280 nm. Purified proteins were aliquoted and stored at −80 °C in PBS (pH 7.4) until use. The production of Nrx1β-Fc from HEK-293 cells and purification on a G protein column was previously described[21]. For production of PSD-95-mcherry, the rat PSD-95 gene was subcloned in frame with a N-terminal mCherry with a 3-fragment strategy into a pET-24a( + ) vector (Novagen) between EcoRI and XhoI restriction sites. Expression of the gene was carried out in E. coli BL21-CodonPlus (DE3)-RIPL with an auto-induction protocol at 16 °C for 20 h. The protein was purified using the C-terminal hexahistidine tag with Ni-NTA resin, dialyzed into PBS containing 0.01% Tween-20, aliquoted and stored at −80 °C until use.

**In vitro kinase screen using Nlg1 gephyrin-binding peptides.** A commercial screen was performed by ProQinase GmbH (Germany), and involved from our side the production and shipping of biotinylated Nlg1 peptides encompassing the 16-aa gephyrin-binding motif (PPDYTLAMRRSPDDVP) that were previously described[25]. A radiometric protein kinase assay (33PanQinase® Activity Assay), based on streptavidin-coated FlashPlate® PLUS plates (PerkinElmer, Boston, MA, USA) was used for measuring the kinase activity of 81 tyrosine kinases. The reaction cocktails were pipetted into 96-well, V-shaped polypropylene microtiter plates ("assay plates") in the following order: 10 μl of kinase solution, and 40 μl of buffer/ATP/test sample mixture. The reaction cocktails contained 60 mM HEPES-NaOH, pH 7.5, 3 mM MgCl$_2$, 3 mM MnCl$_2$, 3 μM Na-orthovanadate, 1.2 mM DTT, 50 μg/ml PEG20000, 1 μM ATP/[γ-33P]-ATP (9.4 × 1005 cpm per well), protein kinase (5–200 ng/50 μl) and sample peptide (1 μM). The assay plate comprised 81 kinase reaction cocktails and one well was used for a peptide/buffer/ substrate control containing no enzyme. The assay plates were incubated at 30 °C for 60 min. Subsequently, reaction cocktails were stopped with 20 μl of 4.7 M NaCl/ 35 mM EDTA. The reaction cocktails were transferred into a 96-well streptavidin-coated FlashPlate® PLUS plate, followed by 30 min incubation at RT on a shaker to allow for binding of the biotinylated peptide to the streptavidin-coated plate surface. Subsequently, the plate was washed three times with 250 μl of 0.9% NaCl. Incorporation of radioactive 33Pi was determined with a microplate scintillation counter (Microbeta, Perkin Elmer). Under the chosen conditions, proteins are binding to the streptavidin-coated FlashPlate® PLUS plates only to very low extent. For evaluation of the results of the FlashPlate® PLUS-based assays, the background signal of each kinase (w/o substrate) had previously been determined in three independent experiments (n = 3) and mean background values (in cpm) had been calculated for each kinase at a radioactivity input of 8.0 × 1005 cpm per well. The

mean (±SD) of all kinase background values was 27 ± 30 cpm. Values above 150 cpm should be considered as significant.

**In vitro kinase assay with GST-Nlg1.** To confirm ourselves the ability of some of those kinases to phosphorylate Nlg1, we used GST-Nlg1 and commercial tyrosine kinases. The in vitro kinase assay was performed by incubating a mix of purified GST-Nlg1 protein with the purified enzyme at 30 °C for 1 h, on a shaker. The mixture consisted of: (1) the purified protein: GST-Nlg1-WT (120 μg/ml) or GST-Nlg1-Y782A (120 μg/ml) or TrkC positive substrate, RBER-NTRK3tide (ProQinase, 25 μg/ml); (2) the enzyme at 1 μg/ml: GST-FGFR1 (V561M) (Sigma-Aldrich,), GST-TrkB or -TrkC (ProQinase GmbH); (3) buffer containing: 60 mM HEPES-NaOH, 3 mM MgCl$_2$, 3 mM MnCl$_2$, 3 μM Na-orthovanadate, 1.2 mM DTT, 0,005% PEG 20000, 50 μM ATP (Sigma, A2383). After incubation, 10 μl of 6× loading buffer (350 mM Tris-HCl, 10% SDS, 30% glycerol, 5% β-mercaptoethanol, 0.06% bromophenol blue, pH = 6.8) was added to each sample, and the mix was heated for 5 min at 60 °C. Samples (13 μl) of the resulting solution were loaded in polyacrylamide gels.

**COS-7 cell culture and transfection.** COS-7 cells (from ATCC) were cultured in DMEM (Eurobio) supplemented with 1% glutamax (GIBCO), 1% sodium pyruvate (Sigma-Aldrich), 10% Fetal Bovine Serum (Eurobio). For GST pull-down, COS-7 cells were plated in 6-well plates (80 000 cells/well) and transfected (using X-tremeGENE™ 9 DNA Transfection Reagent, Roche) with gephyrin-Venus (0.5 μg/well) 36–48 h before the experiment. For Nlg immunoprecipitation (IP), COS-7 cells were plated in 6-well plates at a density of 150,000 cells/well. After 5–6 h, cells were transfected with HA-Nlg1, HA-Nlg2, HA-Nlg3, with or without receptor tyrosine kinases (FGFR1-Flag, TrkB-RFP, HA-TrkC), and left under a humidified 5% CO2 atmosphere (37 °C) for 2 days before being processed for immunoprecipitation.

**Primary neuronal cultures and transfection.** Hippocampal or cortical cultures were prepared from E18 Sprague–Dawley rat embryos (Janvier Labs, Saint-Berthevin, France). Hippocampi and cortex were dissected out in Hibernate medium (for 500 mL MEM, 1 g MOPS, 3.9 mL Glucose 45%, 5 mL Na Pyruvate, pH = 7.3) and subsequently incubated in 5 mL 0.05% trypsin-EDTA, 10 mM HEPES at 37 °C for 20 min. Cells were aspirated up and down for 20 times in a flame-polished Pasteur pipet pre-coated with horse serum. Cell concentration was determined using a Malassez cell counting chamber.

For staining recombinant Nlg1 and endogenous scaffolds, or for single molecule tracking of Nlg1, hippocampal neurons were electroporated with the Amaxa system (Lonza) using 500,000 cells per cuvette. The following plasmid combination was used: GFP, Homer1c-GFP, or gephyrin-Venus + BirA$^{ER}$ + AP-Nlg1-WT, -Y782A, or –Y782F (1.5:1.5:1.5 μg DNA). Electroporated neurons were resuspended in Minimal Essential Medium supplemented with 10% horse serum (MEM-HS) and plated at a density of 5000 cells cm$^{-2}$ on 18-mm coverslips (Marienfeld, 117 580) previously coated for 2 h with 1 mg ml$^{-1}$ polylysine in borate buffer (pH = 8.3). Three hours after plating, coverslips were flipped onto 60-mm dishes containing a glial cell layer in Neurobasal medium supplemented with 2 mM L-glutamine and 1× NeuroCult SM1 Neuronal supplement (STEMCELL Technologies) and cultured for 2 weeks at 37 °C and 5% CO2. Astrocytes were prepared from the same embryos, plated between 20,000 and 40,000 cells per 60-mm dish and cultured in MEM (Fisher Scientific) containing 4.5 g l$^{-1}$ glucose, 2 mM L-glutamine and 10% horse serum (Invitrogen) for up to 14 days before being used as feeder layers for neurons.

For immunostaining of excitatory and inhibitory synaptic markers and electrophysiology experiments, hippocampal neurons were plated at a density of 3000 cells cm$^{-2}$ in Neurobasal medium containing 10% horse serum and 5 μM Ara-C was added at DIV 6–8. Cells were co-transfected at DIV 7 with GFP and either Nlg1-WT, Nlg1-Y782A, Nlg1-Y782F, or EV at a ratio of 1:9, using a lipofection protocol (Effecten, Qiagen), and used for immunocytochemistry or electrophysiology at DIV14–15.

For biochemistry, cortical neurons were plated at a density of 300,000–600,000 cells per well in 6-well plates coated with 1 mg ml$^{-1}$ polylysine. In some experiments, cultures from the same dissection were harvested at different time points (DIV 7–8, 10–11, 14, and 16–17), and solubilized proteins were stored at −80 °C, before being immunoprecipitated and loaded on the same gel after the last time point. In other experiments, tyrosine kinase inhibitors were applied for 24 h to DIV 10–14 cultures prior to cell lysis: 0.5 μM FGFR-inhibitor (PD166866, Sigma), 0.5 μM pan-Trk inhibitor (GNF5837, Tocris), 400 μM EphB2 inhibitor (SNEWIQPRLPQH peptide, thereafter abbreviated SNEW, COV5910, Covalab), 0.5 μM pan-JAK inhibitor I (420099, Calbiochem), or 0.5 μM FLT-3 inhibitor (343020, Calbiochem).

**GST pull-down.** COS-7 cells (ATCC) transfected with gephyrin-Venus were washed with cold PBS, then lysed on ice by adding the following buffer (150 mM NaCl, 50 mM Tris, 1% Triton X-100, 1 mM EDTA, protease inhibitor cocktail III (Calbiochem)) to each well (400 μl/ 6-well plate, 3 wells per condition). Cells were scraped and the collected lysate was incubated on ice for 15 min. After centrifugation (15 min, 8,000 × g, 4 °C) the supernatant was carefully transferred into a

new tube, and kept on ice. In parallel, Pierce® Glutathione Magnetic Beads were equilibrated in a modified PBS solution (PBS pH 7.4, 0.01% BSA, 0.01% Tween-20) and incubated with either GST, GST-Nlg1, GST-Nlg1-Y782A, or GST-Nlg1-Y782F (0.64 nmol/200 μl modified PBS) for 20 min at RT on a shaker. After washing, beads were incubated for another 20 min with the lysate containing Gephyrin-Venus (200 μl/tube), or with purified PSD-95-mCherry (0.01 nmol/100 μl) at RT on a shaker. After washing with modified PBS, the solution containing the coated beads were transferred into new tubes and kept for 10 min on a shaker at room temperature (RT). Beads were thoroughly washed with modified PBS, resuspended into 25 μl Elution Buffer (10 mM glutathione, 50 mM Tris, pH 7.5) and incubated for 10 min at RT on a shaker. Eluates were transferred into new tubes and were heated for 5 min, at 95 °C after adding 5 μl of 6× loading sample buffer to each sample. The resulting solution was loaded onto SDS-PAGE: 17 μl for Coomassie staining and 11 μl for western blotting. After migration, half of the gel was stained with Coomassie blue, while the other half was used for transfer and immunoblotting. After subtraction of non-specific binding to GST, immunoblotted gephyrin or PSD-95 bound to GST-Nlg1 mutant proteins was normalized by anti-GST signal obtained for GST-Nlg1-WT.

**Nlg immunoprecipitation.** COS-7 cells or cortical neurons were treated with 100 μM pervanadate for 15 min before lysis, to preserve phosphate groups on Nlgs. Whole-cell protein extracts were obtained by solubilizing cells in lysis buffer (50 mM HEPES, pH 7.2, 10 mM EDTA, 0.1% SDS, 1% NP-40, 0.5% DOC, 2 mM Na-Vanadate, 35 μM PAO, 48 mM Na-Pyrophosphate, 100 mM NaF, 30 mM phenylphosphate, 50 μM NH$_4$-molybdate and 1 mM ZnCl$_2$) containing protease Inhibitor Cocktail Set III, EDTA-Free (Calbiochem). Lysates were clarified by centrifugation at 8000× g for 15 min. For immunoprecipitations, 500–1000 μg of total protein (estimated by Direct Detect assay, Merck Millipore), were incubated overnight with 2 μg of specific antibodies, then precipitated with 20 μL protein G beads (Dynabeads Protein G, Thermo Fisher Scientific) and washed 4 times with lysis buffer. IP experiments of HA-Nlg1, HA-Nlg2, and HA-Nlg3 were performed with rabbit anti-Nlg1, -Nlg2, or -Nlg3 (Synaptic systems 129013, 129202, and 129113, respectively), or rat anti-HA antibodies (Roche clone 3F10, 11867423001), respectively. At the end of the IP, 20 μL beads were resuspended in 20 μL of 2× loading buffer, and supernatants were processed for SDS-PAGE and western blotting.

**SDS-PAGE and immunoblotting.** Proteins were loaded on 4–15% Mini-PROTEAN TGX Precast Protein Gels, Bio-Rad) for separation (200 V, 400 mA, 40 min) and were afterwards transferred to nitrocellulose membranes for semi-dry immunoblotting (7 min, BioRad). Membranes were rinsed in Tris-buffered saline Tween-20 (TBST; 28 mM Tris, 137 mM NaCl, 0.05% Tween-20, pH 7.4) and incubated with 5% non-fat dried milk for 45 min at RT. Membranes were incubated for 1 h at RT (or overnight at 4 °C) in 0.5% non-fat dried milk in TBST containing the appropriate primary antibodies, as follows: mouse anti-gephyrin (Synaptic Systems clone 3B11, 147111, 1:5000), mouse monoclonal anti-PSD-95 (ThermoFisher Scientific; clone 7E3–1B8, MA1-046, concentration 1:1000), goat anti-GST (GE Healthcare 27-4577-01, concentration: 1:8000), mouse anti-pTyr (1:1000, Cell Signaling 9411 S and Merck 05-1050), rabbit anti-Nlg1, rabbit anti-Nlg2, or rabbit anti-Nlg3 (all at 1:1000, Synaptic systems 129013, 129202, and 129113, respectively), or mouse anti-βIII-tubulin (1:5000, Exbio 11-264-C100). After washing three times with TBST buffer, blots were incubated with horseradish peroxidase (HRP)–conjugated donkey anti-mouse or anti-rabbit secondary antibodies (Jackson Immunoresearch, 715-035-150 and 711-035-152, respectively, concentration: 1:5000) or anti-mouse or anti-rabbit Easyblot (GeneTex, GTX221667-01 or GTX221666-01, respectively, concentration: 1:1000) for IP for 1 h at RT. The latter was used to avoid the detection of primary antibodies from the IP. Target proteins were detected by chemiluminescence with Super signal West Dura or Super signal West Femto (Pierce) or Clarity Western ECL Substrate (Bio-Rad) on the ChemiDoc Touch system (Bio-Rad). Average intensity values were calculated using Image Lab 5.0 software (Bio-Rad). The ratio of phospho-Nlg1/total Nlg1 signal was normalized to the control samples. Uncropped scans of immunoblots from Fig.8 are shown in Supplementary Fig. 11.

**Immunocytochemistry and image analysis of protein clusters.** To visualize surface AP-tagged Nlg1 and endogenous PSD-95 or gephyrin in dissociated neurons, cultures were incubated with Alexa647-conjugated monomeric streptavidin (100 nM) for 10 min at 37 °C and subsequently fixed for 10 min in 4% paraformaldehyde-4% sucrose and permeabilized for 5 min with 0.1% Triton X-100 in PBS. Non-specific binding was blocked with PBS containing 1% Bovine Serum Albumine (BSA). Fixed neurons were immunostained for endogenous PSD-95 or gephyrin using mouse monoclonal anti-PSD-95 (Thermo Fisher Scientific; clone 7E3-1B8, 1:400) or mouse monoclonal anti-gephyrin antibody (Synaptic Systems, clone 3B11, 147111, 1:400) followed by Alexa568-conjugated goat anti-mouse antibody (2 mg ml$^{-1}$, Invitrogen, 1:800).

To stain endogenous AMPA and GABA$_A$ receptors at the cell surface, a 10 min live labeling was performed at 37 °C with a rabbit anti-GluA1 (Agrobio, clone G02141, 0.2 mg ml$^{-1}$, 1:100 dilution in culture medium) or rabbit anti-γ2 antibody (Alomone, AGA-005, 1:100) respectively. This was followed by fixation, membrane

permeabilization and BSA blocking. Cultures were then counterstained for Vglut1 using guinea pig anti-Vglut1 antibody (Merck Millipore, AB5905, 1:2000) or VGAT using guinea pig anti-VGAT (Synaptic Systems, 131004, 1:1000), followed by Alexa568-conjugated goat anti-rabbit or anti-guinea pig antibodies, respectively (2 mg ml$^{-1}$, Molecular Probes, 1:800). Coverslips were mounted in Mowiol (Calbiochem).

For visualization of Nrx1β-Fc bound to transfected neurons, live neurons were incubated for 15 min at 37 °C with Nrx1β-Fc (2 μg/100 μl culture medium) before being fixed. Nrx1β-Fc was then immunostained using a goat anti-human Fc (Jackson Immunoresearch, 1:400) followed by a Alexa568-conjugated donkey anti-goat (Molecular Probes, 1:800). Coverslips were mounted in Mowiol (Calbiochem).

Immunostainings in primary cultures were visualized on an inverted epifluorescence microscope (Nikon Eclipse TiE) equipped with a 60 × /1.40 NA objective and filter sets for EGFP (Excitation: FF01-472/30; Dichroic: FF-495Di02; Emission: FF01-525/30); Alexa568 (Excitation: FF01-543/22; Dichroic: FF-562Di02; Emission: FF01-593/40); and Alexa647 (Excitation: FF02-628/40; Dichroic: FF-660Di02; Emission: FF01-692/40) (SemROCK). Images were acquired with an sCMOS camera (Hamamatsu ORCA Flash 4.0), using the Metamorph® software (Molecular Devices). Detection of Nlg1 clusters was carried out after segmenting the Nlg1 signal using an automatic program written in Metamorph. Nlg1 clusters were considered as containing gephyrin or PSD-95 when ≥20% of the cluster area was positive for thresholded gephyrin or PSD-95 signal. The averaged intensity of gephyrin or PSD-95 signals within individual segmented Nlg1 clusters was measured. The number of VGlut1, VGAT, GluA1, and γ2 clusters per unit dendrite length of GFP-positive neurons was calculated manually in Metamorph.

**Single particle tracking**. Anti-rabbit Fab (Jackson Immunoresearch, 111-007-003) was prepared in PBS at ~1.3 mg ml$^{-1}$. Coupling to the NHS ester derivative of Atto594 was performed following the recommended procedures from the manu-facturer (ATTO-TEC). Briefly, labeling was conducted in the dark at RT for 1 h. Excess dye was removed using Sephadex G-25 medium (PD MiniTrap G-25, GE Healthcare) by elution with PBS. Labeled Fabs were concentrated to ~0.1 mg ml$^{-1}$ using Amicon Ultra centrifugal filters with a 10-kDa cutoff, aliquoted, and flash-frozen for storage at −80 °C until use.

Universal Point Accumulation in Nanoscopic Topography (uPAINT) experiments were carried out as previously reported[28]. Cells were mounted in Tyrode solution (in mM: 15 D-glucose, 108 NaCl, 5 KCl, 2 MgCl$_2$, 2 CaCl$_2$, and 25 HEPES, pH 7.4) containing 1% globulin-free BSA (Sigma) in an open Inox observation chamber (Life Imaging Services, Basel, Switzerland). The chamber was placed on an inverted microscope (Nikon Ti-E Eclipse) equipped with an EMCCD camera (Evolve, Roper Scientific, Evry, France), a thermostatic box (Life Imaging Services) providing air at 37 °C and an APO total internal reflection fluorescence (TIRF) 100 × /1.49 NA oil objective. GFP-expressing cells were detected using a mercury lamp (Nikon Xcite) and appropriate filter sets described above (SemRock). Neurons expressing Homer1c-GFP + Nlg1-WT, Y782A, or Y782F were labeled using a pre-mix containing low concentrations of rabbit anti-GluA1 (Agrobio, clone G02141, 0.2 mg ml$^{-1}$, 1:2000) or rabbit anti-γ2 antibody (Alomone, AGA-005, 1:2000), and Atto594-conjugated anti-rabbit Fab (0.1 mg ml$^{-1}$, 1:2000) to label individual AMPA and GABA$_A$ receptors, respectively. To track biotinylated AP-tagged Nlg1 proteins co-expressed with BirA$^{ER}$, a 1 nM dilution of Atto594-conjugated monomeric streptavidin was used[28]. A four-color laser bench (405; 488; 561; and 642 nm, 100 mW each; Roper Scientific) is connected through an optical fiber to the TIRF illumination arm of the microscope. Laser powers were controlled through acousto-optical tunable filters driven by Metamorph. Atto594 was excited with the 561-nm laser line through a four-band beam splitter (R405/488/561/635, SemRock). Samples were imaged by oblique laser illumination, allowing the excitation of individual Atto-conjugated ligands bound to the cell surface, without illuminating ligands in solution. Fluorescence was collected using a FF01-617/73 nm emission filter (SemRock) placed on a filter wheel (Sutter Instruments). Stacks of 2000 consecutive frames were obtained from each cell, with an integration time of 20–50 ms.

Image stacks were analyzed using a custom program running on Metamorph based on wavelet segmentation for localization and simulated annealing algorithms for tracking, described earlier[66]. The program allows detection and tracking of localized single molecules through successive images. The instantaneous diffusion coefficient, $D$, was calculated for each trajectory containing at least 10 points, from linear fits of the first 4 points of the mean square displacement (MSD) function vs. time. MSD curves were plotted for trajectories containing at least 20 points.

**Organotypic slice culture and single-cell electroporation**. Organotypic hippo-campal slice cultures were prepared as described[67] from either wild type or Nlg1 knockout mice (C57Bl6/J strain) obtained from N. Brose (MPI Goettingen). All animal experiments complied with all relevant ethical regulations (study protocol approved by the Ethical Committee of Bordeaux CE50). Animals were raised in our animal facility; they were handled and euthanized according to European ethical rules. Briefly, animals at postnatal day 5–8 were quickly decapitated and brains placed in ice-cold Gey's balanced salt solution under sterile conditions. Hippocampi were dissected out and coronal slices (350 μm) were cut using a tissue chopper (McIlwain) and incubated at 35 °C with serum-containing

medium on Millicell culture inserts (CM, Millipore). The medium was replaced every 2–3 days. After 3–4 days in culture, slices were transferred to an artificial cerebrospinal fluid (ACSF) containing (in mM): 130 NaCl, 2.5 KCl, 2.2 CaCl$_2$, 1.5 MgCl$_2$, 10 D-glucose, 10 Hepes (pH 7.35, osmolarity adjusted to 300 mOsm). CA1 pyramidal cells were then processed for single-cell electroporation using glass micropipets containing plasmids encoding EGFP (6 ng μl$^{-1}$) along with AP-Nlg1 or EV and BirA$^{ER}$ in equal proportions (13 ng μl$^{-1}$). For knockdown and rescue experiments, a plasmid carrying the Nlg1 specific shRNA (13 ng μl$^{-1}$) was elec-troporated alone or along with a resistant Nlg1 (13 ng μl$^{-1}$). Micropipets were pulled from 1 mm borosilicate capillaries (Harvard Apparatus) with a vertical puller (Narishige). Electroporation was performed by applying 4 square pulses of negative voltage (−2.5 V, 25 ms duration) at 1 Hz, then the pipet was gently removed. A total of 10–20 neurons were electroporated per slice, and the slice was placed back in the incubator for 7 days before electrophysiology or confocal imaging.

**Confocal microscopy and spine counting**. For visualization of recombinant AP-tagged Nlg1 and spine morphology in electroporated CA1 neurons expressing GFP, BirA$^{ER}$, and AP-Nlg1 (WT or mutants), organotypic slices were fixed with 4% paraformaldehyde- 4% sucrose in PBS for 4 h before the permeabilization of membranes with 0.25% Triton in PBS. Slices were subsequently incubated with a monoclonal mouse anti-GFP (Roche, clones 7.1 and 13.1, 1:300) followed by a mixture of Alexa488-conjugated goat anti-mouse antibody (Thermo Fisher Sci-entific, A11001, 1:200) and Alexa647-conjugated Streptavidin (Thermo Fisher Scientific, S21374, 1:200). Images were acquired on a commercial Leica DMI6000 TCS SP5 microscope using a 63 × /1.4 NA oil objective and a pinhole opened to 1 time the Airy disk. Images of 4096 × 4096 pixels, corresponding to a pixel size of 70 nm, were acquired at a scanning frequency of 400 Hz. The vertical step size was set at 0.3 μm. The number of spines per unit dendrite length of GFP-positive cells was calculated manually using Metamorph.

**Electrophysiological recordings**. Whole-cell patch-clamp recordings were carried out from primary hippocampal cultures placed on the stage of a Nikon Eclipse FN1 upright microscope at RT and using Multiclamp 700B amplifier (Axon Instru-ments). The recording chamber was continuously perfused with aCSF containing (in mM): 130 NaCl, 2.5 KCl, 2.2 CaCl$_2$,1.5 MgCl$_2$,10 D-glucose, 10 Hepes, and 0.1 picrotoxin (pH 7.35, osmolarity adjusted to 300 mOsm). The micropipettes were made from borosilicate glass capillaries, with a resistance in the range of 4–6 MΩ. The 2D stage and the micromanipulators were purchased from Scientifica. The internal solution contained (in mM) 135 Cs-MeSO$_4$, 8 CsCl, 10 HEPES, 0.3 EGTA, 4 MgATP, 0.3 NaGTP, and 5 QX-314. mEPSCs recordings were performed in the presence of 0.5 μM TTX. Neurons were held at a membrane potential of −70 mV for mEPSC measurements and at + 10 mV for IPSC measurements. We verified that bicuculine (20 μM) and a combination of CNQX (20 μM) + APV (50 μM) blocked mIPSCs and mEPSCs, respectively. Salts were purchased from Sigma and drugs from Tocris.

To record CA1 neurons in organotypic slices, the chamber was continuously perfused with ACSF bubbled with 95% O$_2$/5% CO$_2$ containing (in mM): 125 NaCl, 2.5 KCl, 26 NaHCO$_3$, 1.25 NaH$_2$PO$_4$, 2 CaCl$_2$, 1 MgCl$_2$, and 25 glucose. CA1 pyramidal neurons were identified with DIC and electroporated neurons were identified by visualizing the GFP fluorescence. EPSCs or IPSCs were evoked in an electroporated neuron and a nearby non-electroporated neuron (control) using a bipolar electrode in borosilicate theta glass filled with ACSF and placed in the stratum radiatum. When recording EPSCs, 20 μM bicuculline was added to block inhibitory synaptic transmission and 100 nM NBQX was added to reduce epileptiform activity. AMPAR- mediated currents were recorded at −70 mV and NMDAR-mediated currents were recorded at +40 mV and measured 50 ms after the stimulus. IPSCs were recorded at +10 mV and in presence of 10 μM NBQX and 50 μM D-AP5 to block AMPARs and NMDARs, respectively. The series resistance Rs was left uncompensated. Recordings with Rs higher than 30 MΩ were discarded. PPR was determined by delivering two pulses separated by 50 ms. PPR was defined as the peak current of the second EPSC over the peak current of the first EPSC. EPSCs and IPSCs amplitudes and PPR measurements were performed using Clampfit (Axon Instruments).

For LTP recordings, ACSF contained in (mM) 125 NaCl, 2.5 KCl, 26 NaHCO$_3$, 1.25 NaH$_2$PO$_4$, 4 CaCl$_2$, 4 MgCl$_2$, and 25 glucose and 0.02 bicuculline and recording pipettes were filled with intracellular solution containing in mM: 125 Cs-MeSO$_4$, 10 CsCl, 10 HEPES, 2.5 MgCl$_2$, 4 Na$_2$ATP, 0.4 NaGTP, and 10 phosphocreatine. Slices were maintained at 25 °C throughout the recording. LTP was induced by depolarization of the cells to 0 mV while stimulating the afferent Schaffer's collaterals at 2 Hz for 100 s.

## Data availability

The authors declare that all data supporting the findings of this study are available within this article, its Supplementary Information files, or are available from the corresponding authors upon reasonable request.

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

## Acknowledgements

We acknowledge A.M. Craig, F. Duschene, F. Janneteau, S. Okabe, P. Scheiffele, A. Ting, and A. Triller for the generous gift of plasmids, N. Brose for the gift of Nlg1 KO mice and Nlg3 plasmid, D. Choquet and C. Breillat for the anti-GluA1 antibody, C. Poujol and S. Marais at the Bordeaux Imaging Center for support in microscopy and image analysis, the animal facility of the University of Bordeaux (in particular A. Lacquemant), the cell culture facility of the Institute (especially S. Benquet, P. Durand and E. Verdier), J. Carrere and R. Sterling for technical assistance. The western blot analysis and recombinant protein production were done in the Biochemistry and Biophysics Platform of the Bordeaux Neurocampus at the Bordeaux University with the help of J.M. Blanc. We thank A. Favereaux and V. Meriel for the quantification of Nlg1 splice variants relative expression in primary hippocampal cultures through qRT-PCR and J. Savas for sharing proteomics data. This work received funding from the Centre National de la Recherche Scientifique, Agence Nationale pour la Recherche (grants « SynAdh » ANR-13-BSV4-0005-01, « SynSpe » ANR-13-PDOC-0012-01, and « Synthesyn » ANR-17-CE16-0028-01), Commission Franco-Américaine (Fulbright program), Conseil Régional Aquitaine (« SiMoDyn »), Investissements d'Avenir Labex BRAIN ANR-10-LABX-43 (« Synapto-genesis »), and Fondation pour la Recherche Médicale (« Equipe FRM » DEQ20160334916), and the national infrastructure France BioImaging (grant ANR-10INBS-04-01).

## Author contributions

O.T. designed the project. M.L., Z.S., I.C., M.S., K.C., and O.T. designed the experiments. M.L., Z.S., I.C., C.S., I.P., B.T., and K.C. performed the experiments and analyzed data. B.T. generated DNA constructs. M.S. provided reagents. M.L. and O.T. wrote the article. M.L., Z.S., I.C., B.T., and O.T. revised the manuscript.

## Additional information

**Competing interests:** The authors declare no competing interests.

