## [Peer Review File · Nature Communications]

Editorial Note: References to figures have been updated at the authors' request to reflect changes made during the revision process. Updates are marked in red.

Reviewers' comments:

Reviewer #1 (Remarks to the Author):

all comments in confidential sections

Reviewer #2 (Remarks to the Author):

The study by Letellier et al., explores the functional and morphological significance of Y782 in neuroligin-1. Endogenous neuroligin-1 is known to be a promoter of excitatory synapse function, however, in vitro NL1 can bind to excitatory and inhibitory scaffolding molecules equally well. Defining the underlying mechanisms that regulate the selective localization and function of NL1 (and other NLs) at excitatory vs inhibitory synapses is critically important for our understanding of these essential molecules and how disease-relevant mutations in NLs might contribute to the etiologies that underlie mental health disorders.

Here, the authors build off of the findings from Giannone et al. (2013) that suggested that phosphorylation of NL1 Y782 biases NL1 to excitatory synapses by preventing Gephyrin binding. Giannone and colleagues made two mutations: 1.) a Y782A mutation that "mimics" phosphorylated Y782, which increases NL1 co-localization with PSD-95 and decreases co-localization with gephyrin; 2. a Y782F is non-phosphorylated version of NL1 that increases co-clustering with gephyrin. In this study, Letellier et al. use an elegant combination of ICC, electrophysiology, optogenetics and single molecule imaging to examine how these two NL1 mutants alter excitatory and inhibitory morphology and basal synaptic transmission and expand on their initial findings. While all versions of NL1 increased excitatory and inhibitory presynaptic densities, Y782A selectively increased GluA1 cluster densities via stabilization of GluA1 and increased mEPSC frequency. Y782F increased gamma-2 cluster density and increased mIPSC frequency. Interestingly, Y782F did not impact AMPAR-mediated synaptic transmission, however, overexpression of this mutant robustly enhanced evoked NMDAR-mediated current. Finally, the authors use a light-activated FGFR1 - a candidate kinase that might phosphorylate NLs in vivo, to increase spine densities in neurons overexpressing WT, but not Y782F NL1.

Overall, this study provides strong evidence regarding NL1 Y782's functional role in controlling excitatory and inhibitory postsynaptic receptor clustering and synaptic transmission, however, the morphological results could have been predicted based on earlier work from the same group (Giannone et al. (2013)). Consistent with their morphological analysis, Y782A selectively enhances excitatory synaptic transmission without a significant change in GABAergic signaling. Their functional finding that Y782F increases excitatory presynapse densities and selectively enhances NMDAR-mediated synaptic transmission is fascinating and largely unexpected. This Y782F finding should be pursued in greater detail because of the well-accepted role for silent synapses in synaptogenesis and synaptic plasticity, and also because it represents a clear advancement from their previous work in 2013. Experiments addressing the comments below would significantly strengthen their conclusions and significance of the study.

Major Comments:

1. One of the most fascinating findings of this paper is in regards to how Y782F selectively increases NMDAR-mediated synaptic transmission AND presynaptic densities. This would strongly suggest that overexpression of unphosphorylated NL1 might promote silent synapse formation. Is this the case or are there more NMDARs per synapse? Taking their morphological data into consideration, the former seems likely. How does the Y782F mutant impact synaptogenesis and what impact does it have on synaptic plasticity? More specifically, does the Y-F mutation increase the number of silent synapses and does this impact LTP?
2. To the best of my knowledge, endogenous NL1 is highly specific for excitatory postsynapses (Song et al., 1999 PNAS) whereas overexpression of NL1 typically displays the 80/20-excitatory/inhibitory localization discussed in this paper. Overexpression experiments are notorious for causing the mislocalization of proteins, however, it has always been remarkable that NL1 still displays such a strong preference for excitatory synapses. One would predict, based on endogenous localization and even overexpression data, that endogenous NL1 would be constitutively phosphorylated in mature neurons under basal conditions. What are the conditions that promote phosphorylation of Y782? Does phosphorylation at this site follow a developmental timecourse or is it activity dependent (e.g. is more unphosphorylated NL1 present early in cultures to help promote silent excitatory synapses followed by an increase in phosphorylation of NL1 to help maintain synapse function)?
3. This work would strongly benefit from more precise experiments that focus on the direct impact of phosphorylation on function. From in vitro binding, the authors identified the Y782A mutation and claim it mimics phosphorylated NL1. This is only true in regards to gephyrin binding - incorporation of the significantly smaller and non-aromatic alanine side chain into NL1 might have other unanticipated consequences. Moreover, in addition to the loss of side chain bulk, the alanine residue lacks the charge that a P-Y would confer.
4. The experiment that addresses tyrosine phosphorylation of NL1 is a technically beautiful experiment, however, as they admit in the discussion, opto-FGFR1 is likely phosphorylating all NLs in addition to other targets. Western blotting looking at the phosphostate of IP'd NL1, NL2 and NL3 following optical activation of FGFR1 from slice or cultured neurons would also be informative.
5. The NL1 cDNA used in this study contains both splice sites A and B. The choice of this isoform was not explained. This isoform binds specifically to beta neurexins without a splice-site 4 insert. Previous overexpression work using this NL1 isoform showed that it only increased excitatory synapse numbers but not synapse size. NL1 without splice site A and B increase synapse density and size (Boucard et al., Neuron. 2005). NL1 Δ AB binds to both alpha and beta neurexins and binding is only slightly modulated by the splice site 4 insert. Neurexin-1a SS4- selectively induced inhibitory synapses in a hemisynapse assay (Chih et al., Neuron, 2006), suggesting that alpha neurexins may predominantly function at inhibitory synapses. More recently, neurexin-3a was shown to govern inhibitory transmission at granule cell – mitral cell synapses in the olfactory bulb. Taken together, would the introduction of Y782F in NL1 Δ AB have: 1. A more pronounced inhibitory synaptic phenotype due to permissive binding to Neurexin-1a? 2. Have a similar impact on inhibitory synapse density and size?

Minor comments:

Figure 1:

1. Spine vs shaft excitatory synapse densities should be quantified, particularly for Y782F. GluA1 cluster size should also be reported. Since Y782F enhances NMDAR-mediated transmission in organotypic slice, do NMDAR cluster densities/sizes change?

Figure 2:

2. VGluT1 and GluA1 cluster sizes should be reported. Localization of VGluT1 (spine vs shaft) should be determined. Does the Y782F increase NMDAR cluster densities/sizes?
3. It is unclear from the methods if the authors are recording from pyramidal or GABAergic neurons in culture. These two classes of cells have very different mEPSC frequencies, amplitudes and mini kinetics. For consistency with organotypic slice electrophysiology, the authors should be recording from pyramidal neurons.

Figure 3:

4. If GluA1 diffusion is slowed by Y782A, why do we not see an increase in mEPSC amplitude? Is it possible to do a similar experiment to look at GABAAR diffusion?

Figure 4:

5. It is surprising that the mIPSC amplitudes are so small (Ave amplitude ~12 pA). With a strong electrochemical driving force of ~80 mV at a holding potential +10 mV, one would expect to see significantly larger mIPSC amplitudes (larger than mEPSC amplitudes) and higher frequencies.

Figure 5:

6. The subpanels F-I are out of order in the text.

7. Are silent synapses generated by O/E of NL1 Y782F? If so, do silent synapses alter the magnitude of LTP?

8. For panels C and D: how were these data acquired? Was an input-output relationship established or do the figures represent data collected from a fixed stimulus intensity? When claiming that AMPAR- or NMDAR-mediated transmission is stronger, it is common to measure the slope of input-output relationship to avoid confounds such as slice variability, # of axons stimulated, etc.

9. Inhibitory transmission should also be explored in organotypic slice.

Figure 7:

10. Why do the authors use NL1 shRNA here instead of the KO used for figure 5? What is the knockdown efficiency?

11. Does opto-FGFR1 activation alone induce spine formation?

12. Do neurons expressing Y782A also display an opto-FGFR1 response in spine numbers?

Discussion:

13. A recent paper by Nguyen et al., 2016. eLife suggest that a mutation analogous to Y782A in NL2 (Y770A) does not alter inhibitory transmission. It would be important to validate the function of this conserved tyrosine residue in NL2 (in a similar way NL3 was tested in this paper) using the same mutations: Y770F and Y770A to measure localization and/or function. At the very least, this paper's findings should be mentioned in the discussion.

Reviewer #3 (Remarks to the Author):

The study by Letellier et al. describes interesting and convincing new findings, extending previous observations of the same group on modifications in the C-terminal domain of Neuroligins (Nlg1) that regulate synaptic specification (Giannone et al., Cell Reports 2013). The study is well-conducted, with a broad array of classical techniques and the addition of state-of-the-art optogenetic manipulation of phosphorylation levels. The study elegantly couples the role of Nlg1 phosphorylation to scaffold association, neurotransmitter receptor recruitment and spine morphology, thereby adding mechanistic insight into how Y782 acts as a unique phosphorylation site in Nlg1 and preferentially assembles functional excitatory synapses over inhibitory synapses.

Understanding the link between neuroligin protein motifs and their function in synaptic assembly is highly relevant, especially in the context of the developmental regulation of the synaptic E/I balance. This study nicely contributes to this topic by addressing molecular details on how neuroligin isoforms with high sequence homology can recruit different postsynaptic machineries by a distinct post-translational modification profile. A criticism is that the experiments rely on overexpressed constructs in cultured neurons, often in a wildtype background. Overall however, the study is well-conducted, well-presented and appropriate for Nat Comm. Several questions remain, and the manuscript would be strengthened if the authors could address the following points.

1) I find it surprising that Nlg1 Y782A has no additional effect over WT Nlg1 in increasing excitatory synapse number, and that Nlg1 Y782F has no additional effect over WT Nlg1 in increasing inhibitory synapses. If the model is that there is normally a (small) pool of WT Nlg1 that is unphosphorylated at Y782 and assembles an inhibitory scaffold via gephyrin, then the Y782A mutant (equivalent to pY782 and incapable of binding gephyrin) would be expected to generate more excitatory synapses than WT. This is not the case in the experiments analysing GluA1 cluster density and mEPSCs, suggesting that WT Nlg1 is already maximally phosphorylated at Y782A. However, if WT Nlg1 is normally maximally phosphorylated at Y782A, then how can it be explained that the Y782F gephyrin-binding mutant does not cause a greater increase in inhibitory synapse formation than WT Nlg1? This suggests to me that describing Y782 as a 'switch' is too strong, and that there might be additional residues in the Nlg1 C-tail or gephyrin-independent pathways. The approach of overexpressing WT and Nlg1 Y782 mutants in a WT background might cause heterodimerization with endogenous Nlg. These points should at least be discussed, but any experimental insight would strengthen the paper.

2) Relevance of Nlg1 Y782 phosphorylation in E/I balance: can the authors provide insight into the phosphorylation state of endogenous Nlg1; does this change over the course of neuronal development and upon manipulation of activity levels? Does increased excitation lead to a Nlg1 dephosphorylation and a shift from excitatory to inhibitory synapses?

3) What is the effect of the Nlg1 $\Delta 5$ and $\Delta 72$ on inhibitory synapse formation? The $\Delta 5$ mutant lacks the PSD-95 binding site but can still interact with gephyrin. Does a Nlg1 Y782F/ $\Delta 5$ double mutant increase inhibitory synapse formation over WT Nlg1 levels? In Fig. 6, the Nlg1 $\Delta 5$ and $\Delta 72$ mutants significantly increase AMPAR EPSCs compared to control despite inability to engage with the postsynaptic density scaffold. As these experiments

were done in Nlg1 KO slices (correct?), does this suggest heterodimerization with Nlg3, or PDZ domain-independent interactions?

4) The Nlg1 Y782F mutant is described as 'blocking' excitatory synapse formation when overexpressed in WT cultured neurons (pg. 8), but when expressed in KO slices still doubles AMPAR EPSC amplitudes compared to controls (Fig. 5C, not significant here). The description of Y782F as 'blocker' is too strong in my opinion. How do the authors explain these different observations?

5) The surface expression levels of Y782F are lower than Y782A based on Nr1 binding (Fig. S2). The decreased colocalization of the Y782F mutant with PSD-95 shows a small decrease that could also be explained by lower surface expression of the mutant. Please comment.

6) What is the effect on PSD-95 binding of Y782A and Y782F? This can be easily tested using the C-tail constructs in Fig. 1C.

7) The section describing the Nlg3 mutants is not clear to me. Did the authors test the effect of the Nlg3 mutations on gephyrin binding? This can be easily tested.

Minor comments:

1) In the intro it is mentioned there are 3 Nlgs in rodents, this should be 4: Nlg1-4.

2) What is the evidence for endogenous Nlg1 at inhibitory synapses? The authors mention a 'sizeable fraction' (pg. 12) at inhibitory synapses and cite two papers (refs 11 and 29), but ref 11 only shows exogenously expressed tagged Nlg1 at inhibitory synapses and ref 29 relies on biotinylating probes fuse to exogenously expressed synaptic adhesion proteins. Is there any evidence for endogenous Nlg1 at inhibitory synapses?

3) Although the authors measure E and I synaptic transmission here in separate experiments (Fig. 2/ Fig. 4); it would be interesting to show E/I ratios of transmission properties measured from the same cell, both in cultured neurons as in the organotypic slices and determine how E/I balance is affected by the different Nlg1 mutants.

4) Please change in text of first results paragraph: "...full length Gephyrin to entire Nlg1", change to C-tail of Nlg1.

5) Please indicate n numbers in figure legends.

6) 1E, G: Please add color code to text left of each panel (B, R, G), this will help to interpret the merged crop for each condition represented.

7) Pg. 10 2nd paragraph: reference to figures with spine density data is wrong: change 5F to 5H; change 5F, G to 5H, I.

8) Pg. 10 3rd paragraph: add reference to commercial screen for tyrosine kinase or show results in supplementary data.

9) The authors should cite Nguyen et al., 2016 eLife, who identify gephyrin-dependent and -independent pathways in the Nlg2 C-tail.

Reference: Letellier et al. "A unique intracellular tyrosine in neuroligin-1 regulates AMPA receptor recruitment during synapse differentiation and potentiation"

Reviewer #1 (Remarks to the Author): all comments in confidential sections
Referee 1 (whose comments were not included) seemed concerned about the effects of the Y782A phosphomimic mutation and validation in neuroligin-2.

We appreciate the point raised by this reviewer, which is most likely motivated by the observation that the gephyrin binding motif containing the critical tyrosine residue is highly conserved among all Nlg isoforms. We thus examined whether Nlg2 and Nlg3 could also be phosphorylated. When expressed in COS cells, both Nlg2 and Nlg3 were phosphorylated by recombinant tyrosine kinases such as TrkB/C (Fig. S8E). In neurons however, only Nlg1 and Nlg3, but not Nlg2, were tyrosine phosphorylated (Fig. 8E,F), potentially reflecting the preferential presence of endogenous tyrosine kinases such as Trks at excitatory synapses. The absence of tyrosine phosphorylation observed for Nlg2 did not encourage us to further study the role of point mutations in Nlg2 (Y770A/F) on inhibitory synaptic transmission. In fact, the Y770A mutation was already shown to inhibit gephyrin binding (Poulopoulos et al., Neuron 2009).

Moreover, compared to Nlg1, the Nlg3 phosphotyrosine signal in neurons was only partially blocked by pan Trk inhibitor (Fig. 8e-g), suggesting that other kinases might phosphorylate an additional tyrosine residue specific to Nlg3 (Y827). Indeed, a Nlg3 construct bearing a Y792A mutation was still significantly phosphorylated compared to Nlg3-WT (Fig. S10a-c). Since Nlg3 is also present at glutamatergic synapses, we investigated whether mutations of the conserved tyrosine residue in Nlg3 (Y792) affected miniature AMPA currents (Fig. S10d-f). The two Nlg3 phosphomutants (Y792F and Y792A), as well as Nlg3-WT, equally increased the frequency of AMPAR-mediated mEPSCs compared to neurons expressing empty vector (Fig. S10d,e). Taken together, these experiments lead us to conclude that the regulation of glutamatergic synapse differentiation by Nlg tyrosine phosphorylation is specific to the Nlg1 isoform.

Reference: Letellier et al. "A unique intracellular tyrosine in neuroligin-1 regulates AMPA receptor recruitment during synapse differentiation and potentiation"

Reviewer #2 (Remarks to the Author):

The study by Letellier et al., explores the functional and morphological significance of Y782 in neuroligin-1. Endogenous neuroligin-1 is known to be a promoter of excitatory synapse function, however, in vitro NL1 can bind to excitatory and inhibitory scaffolding molecules equally well. Defining the underlying mechanisms that regulate the selective localization and function of NL1 (and other NLs) at excitatory vs inhibitory synapses is critically important for our understanding of these essential molecules and how disease-relevant mutations in NLs might contribute to the etiologies that underlie mental health disorders. Here, the authors build off of the findings from Giannone et al. (2013) that suggested that phosphorylation of NL1 Y782 biases NL1 to excitatory synapses by preventing Gephyrin binding. Giannone and colleagues made two mutations: 1.) a Y782A mutation that "mimics" phosphorylated Y782, which increases NL1 co-localization with PSD-95 and decreases co-localization with gephyrin; 2. a Y782F is non-phosphorylated version of NL1 that increases co-clustering with gephyrin. In this study, Letellier et al. use an elegant combination of ICC, electrophysiology, optogenetics and single molecule imaging to examine how these two NL1 mutants alter excitatory and inhibitory morphology and basal synaptic transmission and expand on their initial findings. While all versions of NL1 increased excitatory and inhibitory presynaptic densities, Y782A selectively increased GluA1 cluster densities via stabilization of GluA1 and increased mEPSC frequency. Y782F increased gamma-2 cluster density and increased mIPSC frequency. Interestingly, Y782F did not impact AMPAR-mediated synaptic transmission, however, overexpression of this mutant robustly enhanced evoked NMDAR-mediated current. Finally, the authors use a light-activated FGFR1 - a candidate kinase that might phosphorylate NLs in vivo, to increase spine densities in neurons overexpressing WT, but not Y782F NL1.

Overall, this study provides strong evidence regarding NL1 Y782's functional role in controlling excitatory and inhibitory postsynaptic receptor clustering and synaptic transmission, however, the morphological results could have been predicted based on earlier work from the same group (Giannone et al. (2013). Consistent with their morphological analysis, Y782A selectively enhances excitatory synaptic transmission without a significant change in GABAergic signaling. Their functional finding that Y782F increases excitatory presynapse densities and selectively enhances NMDAR-mediated synaptic transmission is fascinating and largely unexpected. This Y782F finding should be pursued in greater detail because of the well-accepted role for silent synapses in synaptogenesis and synaptic plasticity, and also because it represents a clear advancement from their previous work in 2013. Experiments addressing the comments below would significantly strengthen their conclusions and significance of the study.

We thank the reviewer for these positive comments and constructive criticisms about our work, which allowed us to considerably improve the original manuscript. We have performed many additional experiments involving biochemistry, single molecule tracking, and electrophysiology, in order to clarify the role of Nlg1 tyrosine phosphorylation in excitatory synapse differentiation and potentiation. Compared to the initial version, we have added two entirely new main figures: **Figs. 6 and S5** describe the effects of replacing endogenous Nlg1 by the point mutants Y782A/F on long term potentiation assessed by electrophysiology. **Figs. 8 and S8** address the regulation of Nlg1 phosphorylation by tyrosine kinases using in vitro kinase assays and biochemistry in COS cells and neurons. We have also added single molecule tracking data of Nlg1 point mutants using a labeling strategy based on monomeric streptavidin (**Fig. 1j-k**), and of endogenous GABA-A receptors using an anti-γ2 antibody (**Fig. S6**). Besides, we have measured inhibitory currents in CA1 neurons expressing the Nlg1 Y87A/F mutants or Nlg1-WT (**Fig. S7**). The main text has been reformatted in accordance (changes from the original version appear in blue).

Finally, we have considerably extended our initial data on the optogenetic stimulation of neuroligin-1 phosphorylation, and we feel that the obtained results should now stand alone as a separate study. Indeed, beyond our previous observation that the optogenetic phosphorylation of recombinant Nlg1 increases dendritic spine density (**old Fig. 7 and Fig. S8**), we now show that the phosphorylation of endogenous Nlg1 following light activation of optFGFR1 not only increases spine number but also AMPAR-mediated currents, and that these effects are absent in organotypic slices from Nlg1 KO animals or when endogenous Nlg1 is replaced by the non phosphorylatable Nlg1-Y782F mutant (**see accompanying paper**). This shows that tyrosine phosphorylation of endogenous Nlg1 is sufficient to trigger excitatory synapse differentiation within 24 h.

Given the amount of work and great potential of the approach, we would like to give these results more visibility than just show them in a condensed form at the end of this manuscript. Indeed, the data now represent three main figures, which we would have difficulty to insert into the limits of the Nature

Communications format, and it would be a pity to hide them as supplemental material. Moreover, our new data with tyrosine kinase inhibitors (**Figs. 8 and S8**) show that the FGFR1 receptor is not such a strong candidate to phosphorylate Nlg1 in neurons, compared to Trk family members, so it would be a little misleading to consider the optoFGFR1 other than a tool to acutely manipulate Nlg1 phosphorylation level, independently of endogenous ligand-gated tyrosine kinase receptors. We have formatted this new manuscript as a brief communication and are uploading it here for the reviewers' perusal, since the data may directly answer some of their questions.

Major Comments:

1. One of the most fascinating findings of this paper is in regards to how Y782F selectively increases NMDAR-mediated synaptic transmission AND presynaptic densities. This would strongly suggest that overexpression of unphosphorylated NL1 might promote silent synapse formation. Is this the case or are there more NMDARs per synapse? Taking their morphological data into consideration, the former seems likely. How does the Y782F mutant impact synaptogenesis and what impact does it have on synaptic plasticity? More specifically, does the Y-F mutation increase the number of silent synapses and does this impact LTP?

The measurements of the NMDA/AMPA ratio from our electrophysiological recordings do support the idea that the Nlg1-Y782F mutant fails to recruit AMPARs but not NMDARs in front of glutamatergic terminals, thus effectively assembling new silent synapses. While it is very likely that all three Nlg1 constructs (WT, Y782A and Y782F) increase the amount of NMDARs per synapse when overexpressed in neurons (see for instance Kwon et al., Nat Neuroscience, 2012; Ko et al., EMBO J 2009), our results obtained in both cultured neurons and organotypic slices strongly suggest that the Nlg1-Y782F cannot recruit AMPARs in front of glutamatergic terminals in contrast to Nlg1-WT and -Y782A. To address more directly the question of the reviewer about the NMDAR content per synapse, we performed immunocytochemistry to double label VGLUT1 and GluN1 (see **Figure 1 below**). Unfortunately, we had a hard time to obtain a good immunosignal for GluN1 using a commercial antibody (Synaptic Systems, clone M68, cat n°114011). While the live labeling prior fixation with methanol or PFA 4% to visualize surface receptors did not give any specific immuno-signal, the post-fixation labeling was very unreliable and thereby did not provide precise information about the NMDARs content at the postsynaptic membrane.

Figure 1: Immunostaining of GluN1, VGlut1 on cultured hippocampal neurons at DIV 14. Neurons were transfected at DIV 7 with homer1c-GFP as a synaptic marker and fixed at DIV 14 in methanol at -20°C for 10min prior immunolabeling. Note the absence of synaptic enrichment of GluN1 immunosignal.

To investigate the effect of the mutants expression on synaptic plasticity, we performed additional experiments consisting of inducing long term potentiation (LTP) in CA1 neurons expressing the two Nlg1 point mutants (Y782A/F) or Nlg1-WT, in a near-replacement configuration (**Fig. 6A-E**). Since LTP is reduced in Nlg1 KO mice, we used organotypic hippocampal slices from Nlg1 WT mice, so that control unelectroporated neurons would show normal LTP. To reduce the effects of Nlg1 over-expression that could also alter LTP, we co-electroporated CA1 neurons with AP-tagged Nlg1-WT, -Y782A, or -Y782F with shRNA to Nlg1 containing a GFP reporter, and BirA^{ER} for subsequent labeling with streptavidin. Compared with non-electroporated neurons, basal AMPAR-mediated EPSCs were about 100% for Nlg1-WT, 400% for Nlg1-Y782A, and 100% for Nlg1-Y782F, suggesting that under replacement conditions, these two Nlg1 mutants differently retain AMPARs (**Fig. 6F**). Using a pairing protocol to elicit LTP, control unelectroporated neurons showed a 2.5-fold increase in AMPAR-mediated EPSCs during at least 30 min. Neurons expressing Nlg1-WT exhibited normal LTP compared to non-electroporated counterparts (**Fig. S5**), validating the Nlg1 replacement strategy. In contrast, neurons expressing Nlg1-Y782F displayed a significantly reduced LTP, in accordance with their difficulty in recruiting PSD scaffolds and AMPARs (**Fig. 6G,H**). In other words, as anticipated by this reviewer, the Y782F mutant favors silent synapses, that are unable to recruit AMPARs upon LTP induction. Strikingly, neurons expressing Nlg1-Y782A did not show any LTP, consistent with the capacity of this Nlg1 mutant to recruit AMPARs in a constitutive manner, thereby blocking further potentiation. Thus, the Y782 residue is not only important for basal synaptic transmission, but also for synaptic plasticity.

2. To the best of my knowledge, endogenous NL1 is highly specific for excitatory postsynapses (Song et al., 1999 PNAS) whereas overexpression of NL1 typically displays the 80/20-excitatory/inhibitory localization discussed in this paper. Overexpression experiments are notorious for causing the mislocalization of proteins, however, it has always been remarkable that NL1 still displays such a strong preference for excitatory synapses. One would predict, based on endogenous localization and even overexpression data, that endogenous NL1 would be constitutively phosphorylated in mature neurons under basal conditions. What are the conditions that promote phosphorylation of Y782? Does phosphorylation at this site follow a developmental time course or is it activity dependent (e.g. is more unphosphorylated NL1 present early in cultures to help promote silent excitatory synapses followed by an increase in phosphorylation of NL1 to help maintain synapse function)?

To address this point, we examined the regulation of the endogenous level of Nlg1 tyrosine phosphorylation in cultured neurons, by performing Nlg1 immunoprecipitation followed by phosphotyrosine (pTyr) immunoblots. We first examined the Nlg1 pTyr level at different time points in culture (DIV 7, 10, 14, and 17) (**Fig. S8A,B**). The P-Tyr level, normalized by the level of Nlg1, seems higher around DIV 10, a period of active synaptogenesis. This suggests that a signal originating during synaptogenesis is triggering Nlg1 phosphorylation, e.g. the binding to NRXs upon the formation of axon/dendrite contacts. In support of this concept, the tyrosine kinases TrkB and Trk C were found in a proteomics screen of pan-NRX interactors (**Fig. S9**). Second, we tested the effect of several tyrosine kinase inhibitors on the pTyr level of endogenous Nlg1 in neurons, after validating the activity of those inhibitors in COS cells co-expressing Nlg1 and recombinant tyrosine kinases (FGFR1, TrkB, TrkC) (**Fig. 8C,D**). Those tyrosine kinase candidates were chosen from an initial screen relying on an in vitro kinase assay (**Fig. S8C,D and Fig. 8A,B**). We show that, although most inhibitors reduce Nlg1 pTyr levels in neurons to some degree (15-50%), the pan-Trk inhibitor (GNF5823) almost entirely blocks the Nlg1 pTyr level (**Fig. 8E,G**). Encouraged by this result, we tested the effect of GNF5823 on synaptic transmission in organotypic hippocampal slices. The inhibitor totally blocked the increase of AMPAR-mediated EPSCs caused by the expression of Nlg1-WT in CA1 neurons from the Nlg1 KO background (**Fig. 8H,I**). In contrast, the FGFR1 inhibitor caused a modest reduction of Nlg1 P-Tyr level, and did not prevent the increase in synaptic transmission caused by Nlg1 expression (**Fig. S9**). Together, these new results indicate that the Nlg1 tyrosine phosphorylation level in neurons is significant and can be regulated over developmental age, most likely through the Trk family of tyrosine kinases. In parallel, the optoFGFR1 experiments suggest that endogenous Nlg1 is not maximally (constitutively) phosphorylated, because we can induce major effects on dendritic spines and AMPAR currents by triggering its phosphorylation (**see accompanying manuscript**). Therefore, our conclusion from these experiments is that Nlg1 PTyr level is regulated in neurons, most likely by a balance between phosphorylation via tyrosine kinases such as Trks, and dephosphorylation by phosphatases which remain to be identified. This balance finely regulates the amount of AMPARs that can be recruited by the associated PSD scaffold at excitatory synapses.

3. This work would strongly benefit from more precise experiments that focus on the direct impact of phosphorylation on function. From in vitro binding, the authors identified the Y782A mutation and claim it

mimics phosphorylated NL1. This is only true in regards to gephyrin binding - incorporation of the significantly smaller and non-aromatic alanine side chain into NL1 might have other unanticipated consequences. Moreover, in addition to the loss of side chain bulk, the alanine residue lacks the charge that a P-Y would confer.

We agree with the reviewer that the Y782A mutation only mimics the effect of phosphorylated NLG1 in terms of gephyrin binding. To directly manipulate the tyrosine phosphorylation of endogenous Nlg1, we first used a pan-Trk inhibitor (GNF), which almost completely blocked Nlg1 pTyr level, and the increase of AMPAR-mediated EPSCs caused by the expression of Nlg1-WT in CA1 neurons (**Fig. 8E-I**). Importantly, neurons expressing Nlg1-Y782A were resistant to the inhibitor, demonstrating that the increase of synaptic responses caused by Nlg1 expression involves Y782 phosphorylation by Trks. This result demonstrates that Nlg1 phosphorylation is important to mediate its synaptogenic function.

In parallel, triggering Nlg1 tyrosine phosphorylation by light-activating the optoFGFR1 increased dendritic spine numbers and AMPAR-mediated EPSCs, but not NMDAR-mediated EPSCs (**see accompanying manuscript**), showing selectivity of Nlg1 phosphorylation on the enhancement of AMPAR-dependent synaptic transmission. Importantly, the enhancement of AMPA currents and spine density were not seen in neurons from Nlg1 KO mice, demonstrating that Nlg1 phosphorylation is selectively mediating these effects. Together, our data show bi-directional effects of Nlg1 tyrosine phosphorylation on synaptic morphology and function.

4. The experiment that addresses tyrosine phosphorylation of NL1 is a technically beautiful experiment, however, as they admit in the discussion, opto-FGFR1 is likely phosphorylating all NLs in addition to other targets. Western blotting looking at the phosphostate of IP'd NL1, NL2 and NL3 following optical activation of FGFR1 from slice or cultured neurons would also be informative.

We thank the reviewer for raising this interesting point. As expected from the consensus sequence in the gephyrin binding motif shared by all NLGs, one would suspect that Nlg2 and Nlg3 could also be phosphorylated by optoFGFR1. In response to this comment, we have examined whether Nlg2 and Nlg3 could also be phosphorylated. When expressed in COS cells, both Nlg2 and Nlg3 were phosphorylated by recombinant tyrosine kinases such as TrkB/C (**Fig. S8E**). In neurons however, only Nlg1 and Nlg3, but not Nlg2, were tyrosine phosphorylated (**Fig. 8F,G**), potentially reflecting the preferential presence of endogenous tyrosine kinases such as Trks at excitatory synapses. In addition, one might expect that the phosphorylation of endogenous NLG2 by optoFGFR1 might prevent NLG2 binding to gephyrin, thereby affecting inhibitory synaptic transmission. However, inhibitory synaptic transmission was not altered upon light activation of optoFGFR1 (**accompanying manuscript, Fig. S1**). Furthermore, the effects of optoFGFR1 on excitatory synapse morphology and function were abolished in the Nlg1 KO background, showing that they are selective for Nlg1 (**accompanying manuscript, Fig. 2**). Thus, despite the fact that the optoFGFR1 might have other targets, it affects excitatory synaptic differentiation through Nlg1 phosphorylation.

Regarding Nlg3, the Nlg3 pTyr signal in neurons was only partially blocked by the pan-Trk inhibitor, suggesting that other kinases might phosphorylate an additional tyrosine residue specific to Nlg3 (Y827). Indeed, a Nlg3 bearing only the Y792A mutation was still significantly phosphorylated compared to Nlg3-WT (**Fig. S10a-c**). Since Nlg3 is also present at glutamatergic synapses, we investigated whether mutations of the conserved tyrosine residue in Nlg3 (Y792) affected miniature AMPA currents. The two Nlg3 phosphomutants (Y792F and Y792A), as well as Nlg3-WT, equally increased the frequency of AMPAR-mediated mEPSCs compared to neurons expressing empty vector (**Fig. S10d-e**). Taken together, these experiments lead us to conclude that the regulation of glutamatergic synapse differentiation by Nlg tyrosine phosphorylation is specific to the Nlg1 isoform.

5. The NL1 cDNA used in this study contains both splice sites A and B. The choice of this isoform was not explained. This isoform binds specifically to beta neurexins without a splice-site 4 insert. Previous overexpression work using this NL1 isoform showed that it only increased excitatory synapse numbers but not synapse size. NL1 without splice site A and B increase synapse density and size (Boucard et al., *Neuron*, 2005). NL1ΔAB binds to both alpha and beta neurexins and binding is only slightly modulated by the splice site 4 insert. Neurexin-1a SS4- selectively induced inhibitory synapses in a hemisynapse assay (Chih et al., *Neuron*, 2006), suggesting that alpha neurexins may predominantly function at inhibitory synapses. More recently, neurexin-3a was shown to govern inhibitory transmission at granule cell – mitral cell synapses in the olfactory bulb. Taken together, would the introduction of Y782F in NL1ΔAB have: 1. A more pronounced inhibitory synaptic phenotype due to permissive binding to Neurexin-1a? 2. Have a similar impact on inhibitory synapse density and size?

We thank the reviewer for this very interesting comment. We have used the Nlg1 splice variant containing both splice sites A and B, because it is the most abundant form of endogenous Nlg1 in the hippocampus, at the mRNA level (Chih et al., Neuron 2006). To confirm that in our experiments Nlg1A/B was indeed the most abundant splice variant, we performed q-RT-PCR of RNAs extracted from DIV 14 hippocampal cultures. We generated eight DNA primers allowing the identification of the four NLG1 splice variants Nlg1AB, Nlg1A, Nlg1B, and Nlg1(-), that were validated in COS-7 cells expressing each of those splice variants. Analysis of mRNAs from cultured hippocampal neurons showed that Nlg1AB is indeed the most abundant form ($\geq 80\%$), followed by NLG1B ($\leq 20\%$), but that NLG1 lacking the splice insert B, i.e. Nlg1A and Nlg1(-), are almost undetectable at the transcript level (Figure 2 below). This actually raises the question of the validity of expressing Nlg1 splice variants that are not present endogenously, unless their expression is somehow regulated. Anyway, following the reviewer's suggestion, and given that the B site insert seems to be the master switch regulating binding to excitatory versus inhibitory pre-synapses, we have expressed the Nlg1A splice variant in CA1 neurons from NLG1 KO slices and measured dendritic spine numbers by confocal microscopy as well as AMPAR- and NMDAR-mediated currents by electrophysiology. Neurons expressing Nlg1A did not display an increase in dendritic spine density as compared to neurons expressing Nlg1-AB, and showed modest increases in AMPAR- and NMDAR-mediated EPSCs (**Figure 2 below**). Given the very small phenotype of the NLG1A splice variant on excitatory synapse differentiation and the extremely low mRNA expression level, we felt that introducing a Y782F mutation in the Nlg1A splice variant was not going to bring a lot of information on how tyrosine phosphorylation of endogenous Nlg1 affects synapse differentiation. Finally, in dissociated hippocampal cultures, the over-expression of Nlg1A caused a strange "filopodial" phenotype (not shown), and in CA1 neurons, the accumulation of Nlg1A in aberrant locations outside spines, so overall we feel that this splice variant is somehow non-physiologic.

Figure 2: Nlg1AB expression predominates in hippocampal neurons and Nlg1 lacking the splice site B has modest synaptogenic properties. (A) qRT-PCR results showing the relative expression of Nlg1 mRNAs containing the extracellular splice sites A (left)

or B (right). **(B)** Confocal images showing dendrites from CA1 neurons electroporated with tdTomato (red), BirA^{ER} and recombinant Nlg1 (live stained with monomeric streptavidin, green). On the right, quantification of spine density. **(C)** AMPAR- and NMDAR-mediated EPSCs recorded from electroporated CA1 neurons in the dual patch-clamp paradigm. Graphs on the right show quantification of the respective amplitudes.

Minor comments:

Figure 1:

1. Spine vs shaft excitatory synapse densities should be quantified, particularly for Y782F. GluA1 cluster size should also be reported. Since Y782F enhances NMDAR-mediated transmission in organotypic slice, do NMDAR cluster densities/sizes change?

Figure 2:

2. VGluT1 and GluA1 cluster sizes should be reported. Localization of VGluT1 (spine vs shaft) should be determined. Does the Y782F increase NMDAR cluster densities/sizes?

We agree with the reviewer that analyzing the fine subcellular distribution of glutamatergic synapses should provide useful information about whether Ngl1-Y782F assembles silent synapses directly on the shaft as we could expect. Indeed, this would explain the effect of this mutant on both the spine density and the increase in NMDA/AMPA ratio. However, we find such analysis extremely difficult to carry out using epifluorescence images showing limited spatial resolution. In addition, spine morphologies and densities can vary quite a lot across cultured neurons and cultures, in particular upon Nlg1 expression.

Also, we considered that the quantification of GluA1 cluster size was not absolutely required given that our electrophysiology data clearly showed no change in the amplitude of the AMPAR-mediated mEPSCs. This is in agreement with previous reports that Nlg1 overexpression enhances the number of functional synapses without affecting their content in PSD-95 or AMPARs (see for instance, Prange et al., PNAS, 2004; Levinson et al., J Biol Chem 2005; Kwon et al., Nature Neurosci, 2012). Although we did not quantify the VGluT1 cluster size in the present study either, several reports from different labs previously showed that the presynaptic size measured from VGluT1 or synapsin immunostainings is increased when overexpressing Nlg1, through a mechanism requiring the binding to presynaptic neurexins (see for instance Ko et al., EMBO J, 2009; Prange et al., PNAS, 2004; Chanda et al., J Neurosci, 2017). However, despite these presynaptic changes that are likely to occur in our culture system as well, we did not find any significant change in the short-term plasticity (PPR) measured in slices, suggesting that presynaptic release of glutamate is not affected in our conditions. Finally, as mentioned above (response to major comment 1), we were unable to obtain good and reliable immunolabelling of GluN1 in our conditions.

3. It is unclear from the methods if the authors are recording from pyramidal or GABAergic neurons in culture. These two classes of cells have very different mEPSC frequencies, amplitudes and mini kinetics. For consistency with organotypic slice electrophysiology, the authors should be recording from pyramidal neurons.

Although it is always possible that some of the recorded neurons were indeed GABAergic neurons, we are confident that most of them were glutamatergic for the following two reasons:

- In our culture conditions, GABAergic neurons represent only about 20% of the total population of neurons.
- We targeted neurons with a relatively large cell body displaying a pyramidal shape with a few main dendrites emerging from it (DIC or GFP signal). From our previous experience using pair recordings, such neurons are indeed glutamatergic as excitatory currents can be elicited in their postsynaptic partners when stimulating them (see Letellier et al., PNAS, 2016; Vitureira et al., Nature Neuroscience, 2011).

Figure 3:

4. If GluA1 diffusion is slowed by Y782A, why do we not see an increase in mEPSC amplitude? Is it possible to a similar experiment to look at GABAAR diffusion?

We measured here the global diffusion coefficient of AMPARs (pooling trajectories both outside and inside synapses). This parameter drops when the number of synapses is increased, which occurs during synaptogenesis or upon Nlg1 expression (see Czöndör et al., PNAS 2012 for the experimental and predicted relationship between synapse number and global AMPAR diffusion). The AMPAR diffusion coefficient inside synapses does not decrease upon Nlg1 expression, so the mEPSC amplitude is not expected to be affected by Nlg1 expression, as previously reported (Levinson et al., JBC 2005).

In response to the reviewer's comment, we performed SPT experiments to track GABA-A receptors at the cell surface using anti- $\gamma 2$ antibodies in combination with Atto-647 conjugated anti-rabbit Fabs (same labeling strategy as for AMPARs). The data were collected in dissociated neurons co-expressing gephyrin-mRFP + either empty vector, Nlg1-WT, Nlg1-Y782A, or Nlg1-Y782F. Results show that both Nlg1-WT and Nlg1-Y782A slightly decrease GABA_A receptor diffusion, while Nlg1-Y782F decreases it more (**Fig. S6**). These findings are consistent with the strong recruitment of gephyrin and GABA_A receptors by the Nlg1-Y782F mutant.

Figure 4:

5. It is surprising that the mIPSC amplitudes are so small (Ave amplitude ~12 pA). With a strong electrochemical driving force of ~80 mV at a holding potential +10 mV, one would expect to see significantly larger mIPSC amplitudes (larger than mEPSC amplitudes) and higher frequencies.

We agree that the mIPSC amplitudes are surprisingly small, but they are in the same order of magnitude as those already reported in the literature using the same types of recordings in hippocampal cultures (Prange et al. PNAS 2004; Levinson et al. JBC 2005). It may be due to the combination of intracellular and extracellular recording solutions which allow the measurement of both mEPSCs and mIPSCs sequentially, upon changing the holding potential. A potential explanation for these low mIPSCs amplitudes is that GABA-A receptors may desensitize at positive voltages thus causing an inward rectification of the I/V curve (Pytel et al., Neuropharmacology 2006), although this is not clear whether this holds true in conditions of synaptic transmission.

Figure 5:

6. The subpanels F-I are out of order in the text.

This was corrected.

7. Are silent synapses generated by O/E of NL1 Y782F? If so, do silent synapses alter the magnitude of LTP?

Please see our response to the major comment 1.

8. For panels C and D: how were these data acquired? Was an input-output relationship established or do the figures represent data collected from a fixed stimulus intensity? When claiming that AMPAR- or NMDAR-mediated transmission is stronger, it is common to measure the slope of input-output relationship to avoid confounds such as slice variability, # of axons stimulated, etc.

The dual patch-clamp data are collected from a fixed intensity stimulus, which is adjusted to produce roughly a constant response (typically around 20-50 pA) in the non-electroporated neighboring neuron which serves as a paired control. The rationale is that the two neurons (electroporated and non-electroporated) are contacted by roughly the same number of stimulated axons. Indeed, when EPSCs are measured in a neuron expressing GFP + empty vector, they are of similar amplitude to those evoked in the non-electroporated neuron, thus validating the experimental design. All measurements are normalized to those obtained in parallel for the control neuron (individual data are plotted in **Figs. S3, S7, and S9**). This normalization procedure has been widely used (Shipman et al., 2011; N'Guyen et al., 2017), and bypasses the need to establish an input/output relationship when doing single patch-clamp recordings. It is also very important for LTP experiments, since the non-electroporated neuron serves as a control that the LTP protocol works (**Figs. 6 and S5**).

9. Inhibitory transmission should also be explored in organotypic slices.

In response to the reviewer's comment, we have performed electrophysiological recordings of inhibitory synaptic transmission in CA1 neurons expressing Nlg1-WT, -Y782A, or -Y782F slices in organotypic slices from Nlg1 KO mice (**Fig. S7**). Surprisingly, all Nlg1 constructs induced a similar ~5-fold enhancement of IPSCs compared to non-electroporated neighboring neurons, independently of the mutation, while based on our results in dissociated cultures, we were expecting Nlg1-Y782A to prevent this increase (**Fig. 7**). These results indicate that the Nlg1 tyrosine point mutants act differently on inhibitory synapse differentiation in organotypic slices and dissociated cultures. One explanation for this discrepancy might stem from the fact that the recorded IPSCs are of different origin in the two systems. In organotypic slices, pyramidal CA1 neurons are connected by two types of interneurons: parvalbumin (PV) interneurons which form synapses mostly on the soma and proximal dendrites, and somatostatin (SOM) interneurons which target essentially distal dendrites. A recent

paper using optogenetics to stimulate either SOM or PV interneurons shows that the SOM but not the PV inputs, are highly sensitive to the Nlg3 expression level, while both inputs are affected by NLG2 or gephyrin KD in CA1 neurons (Horn and Nicoll, PNAS 2017). By electrical stimulation in slices, we likely trigger PV interneurons, which have the most abundant and strongest synaptic inputs onto CA1 neurons. One explanation for the fact that the expression of NLG1-Y782A strongly increases IPSCs is that it may dimerize with endogenous NLG2, which is still able to connect to gephyrin via its proline-rich collybistin-binding motif, resulting in an increase of evoked IPSCs. In cultures, the inhibitory inputs are primarily made on dendrites (**Fig. 3 below**), where Nlg3 likely play an important and selective role (Horn and Nicoll, PNAS 2017). We suspect that Nlg1-Y782A acts as a dominant negative construct by forming heterodimers with endogenous Nlg3, thus preventing gephyrin recruitment (Nlg3 lacks the collybistin-binding motif which is specific of Nlg2). Hence the inability of Nlg1-Y782A to increase the frequency of mIPSCs in dissociated cultures.

Figure 3: Epifluorescence images showing immunolabelling for endogenous VGAT (red), gephyrin (green). Expression of GFP (blue) allows us to visualize neuron morphology. Note that many synapses are found on distal dendrites, and no so many on the cell body.

Figure 7:

10. Why do the authors use NL1 shRNA here instead of the KO used for figure 5? What is the knockdown efficiency?

We used a knock-down of NLG1 instead of a KO when we wanted to have a control non-electroporated cell containing endogenous Nlg1 for comparison. This was especially critical for LTP experiments because LTP is impaired in Nlg1 KO neurons (Blundel et al., 2010). We also used this approach for the optoFGFR1 experiments because we wanted that the levels of recombinant NLG1 mutants roughly matched those of endogenous NLG1 expressed by neighbouring cells, in order to prevent any competition that might takes place between cells expressing different levels of Nlg1 (see Kwon et al., Nature Neuroscience, 2012). The efficiency of the KD was previously measured in primary cultures by immunoblot using an antibody to endogenous NLG1, and by surface labeling of endogenous NLG1 with purified NRX1 β -Fc, and is around ~70 % (Chamma et al., Nat Comms 2016).

11. Does opto-FGFR1 activation alone induce spine formation?

Yes it does. We have now expressed opto-FGFR1 +Td Tomato in CA1 neurons from NLG1 WT vs Nlg1 KO slices. In the NLG1 WT background, light stimulation induces an increase in dendritic spine density and in AMPAR-mediated EPSCs, which is not found in the NLG1 KO background, demonstrating that NLG1 tyrosine phosphorylation is mediating these effects (see accompanying paper Fig. 2). These experiments also show that endogenous NLG1 is not maximally phosphorylated in neurons because we can induce major effects by triggering its phosphorylation. Please note however that we would now like to include those data in a separate manuscript.

12. Do neurons expressing Y782A also display an opto-FGFR1 response in spine numbers?

Given our new data showing that the effects of optoFGFR1 on spine numbers and AMPAR-mediated EPSCs are not found in the NLG1 KO background, we feel that this additional control is not essential any longer. We initially chose the Y782F mutant as a control that cannot be phosphorylated, the phenylalanine being a structurally better mimic of unphosphorylated tyrosine than alanine. Instead, we now show that, while the pan-Trk inhibitor GNF5823 blocks the increase in synaptic transmission induced by Nlg1 WT, the Nlg1-Y782A mutant is resistant to the inhibitor, showing that the tyrosine residue Y782 in Nlg1 is a target of the Trk family of tyrosine kinases.

Discussion:

13. A recent paper by Nguyen et al., 2016. eLife suggest that a mutation analogous to Y782A in NL2 (Y770A) does not alter inhibitory transmission. It would be important to validate the function of this conserved tyrosine residue in NL2 (in a similar way NL3 was tested in this paper) using the same mutations: Y770F and Y770A to measure localization and/or function. At the very least, this paper's findings should be mentioned in the discussion.

We thank the reviewer for mentioning this paper which we originally omitted to cite (now done in the discussion). As explained in our response to point n°4, we performed additional experiments showing that although recombinant Nlg2 can also be phosphorylated by tyrosine kinases such as TrkB/C (Fig. S8E), endogenous Nlg2 was not tyrosine phosphorylated in neurons. We feel therefore that the Y770A/F mutation should not make such a strong contribution as for Nlg1, especially considering the selective collybistin-binding motif downstream of the transmembrane domain present only in Nlg2 (proline rich region), and potentially the specificity of the Nlg2 extracellular domain. Thus, we were not strongly encouraged to measure IPSCs in neurons expressing Nlg2-Y770A/F mutants. Furthermore, the fact that activation of the optoFGFR1 does not affect inhibitory currents argues against a major role of Nlg2 phosphorylation on inhibitory synaptic transmission (**accompanying paper, Fig. S1**). Together with the observation that the Nlg3-Y892F mutant is still able to increase mEPSCs (**Fig. S11**), our data thus seem to show that the tyrosine phosphorylation pathway is primarily regulating the function of the Nlg1 isoform.

Reference: Letellier et al. "A unique intracellular tyrosine in neuroligin-1 regulates AMPA receptor recruitment during synapse differentiation and potentiation"

Reviewer #3 (Remarks to the Author):

The study by Letellier et al. describes interesting and convincing new findings, extending previous observations of the same group on modifications in the C-terminal domain of Neuroligins (NLGs) that regulate synaptic specification (Giannone et al., Cell Reports 2013). The study is well-conducted, with a broad array of classical techniques and the addition of state-of-the-art optogenetic manipulation of phosphorylation levels. The study elegantly couples the role of Nlg1 phosphorylation to scaffold association, neurotransmitter receptor recruitment and spine morphology, thereby adding mechanistic insight into how Y782 acts as a unique phosphorylation site in Nlg1 and preferentially assembles functional excitatory synapses over inhibitory synapses. Understanding the link between neuroligin protein motifs and their function in synaptic assembly is highly relevant, especially in the context of the developmental regulation of the synaptic E/I balance. This study nicely contributes to this topic by addressing molecular details on how neuroligin isoforms with high sequence homology can recruit different postsynaptic machineries by a distinct post-translational modification profile.

We thank the reviewer for these positive comments and constructive criticisms about our work, which allowed us to considerably improve the original manuscript. We have performed many additional experiments involving biochemistry, single molecule tracking, and electrophysiology, in order to clarify the role of Nlg1 tyrosine phosphorylation in excitatory synapse differentiation and potentiation. Compared to the initial version, we have added two entirely new main figures: **Figs. 6 and S5** describe the effects of replacing endogenous Nlg1 by the point mutants Y782A/F on long term potentiation assessed by electrophysiology. **Figs. 8 and S8** address the regulation of Nlg1 phosphorylation by tyrosine kinases using in vitro kinase assays and biochemistry in COS cells and neurons. We have also added single molecule tracking data of Nlg1 point mutants using a labeling strategy based on monomeric streptavidin (**Fig. 11,J,K**), and of endogenous GABA-A receptors using an anti- γ 2 antibody (**Fig. S6**). Besides, we have measured inhibitory currents in CA1 neurons expressing the Nlg1 Y87A/F mutants or Nlg1-WT (**Fig. S7**). The main text has been reformatted in accordance (changes from the original version appear in blue).

Finally, we have considerably extended our initial data on the optogenetic stimulation of neuroligin-1 phosphorylation, and we feel that the obtained results should now stand alone as a separate study. Indeed, beyond our previous observation that the optogenetic phosphorylation of recombinant Nlg1 increases dendritic spine density (**old Fig. 7 and Fig. S8**), we now show that the phosphorylation of endogenous Nlg1 following light activation of optFGFR1 not only increases spine number but also AMPAR-mediated currents, and that these effects are absent in organotypic slices from Nlg1 KO animals or when endogenous Nlg1 is replaced by the non phosphorylatable Nlg1-Y782F mutant (**see accompanying paper**). This shows that tyrosine phosphorylation of endogenous Nlg1 is sufficient to trigger excitatory synapse differentiation within 24 h.

Given the amount of work and great potential of the approach, we would like to give these results more visibility than just show them in a condensed form at the end of this manuscript. Indeed, the data now represent three main figures, which we would have difficulty to insert into the limits of the Nature Communications format, and it would be a pity to hide them as supplemental material. Moreover, our new data with tyrosine kinase inhibitors (**Figs. 8 and S8**) show that the FGFR1 receptor is not such a strong candidate to phosphorylate Nlg1 in neurons, compared to Trk family members, so it would be a little misleading to consider the optoFGFR1 other than a tool to acutely manipulate Nlg1 phosphorylation level, independently of endogenous ligand-gated tyrosine kinase receptors. We have formatted this new manuscript as a brief communication and are uploading it here for the reviewers' perusal, since the data may directly answer some of their questions.

A criticism is that the experiments rely on overexpressed constructs in cultured neurons, often in a wildtype background. Overall however, the study is well-conducted, well-presented and appropriate for Nat Comm. Several questions remain, and the manuscript would be strengthened if the authors could address the following points.

We agree with the reviewer remark that all of our experiments in primary neurons were done by over-expressing NLG1 mutants in a wild type background (**Figs 1-3 and 7**). However, most experiments performed in organotypic slices (**Figs. 4, 5 and 8**) were performed in a Nlg1 KO background, where recombinant Nlg1 mutants cannot homodimerize with endogenous Nlg1 (although they may form heterodimers with Nlg3).

Finally, LTP experiments in Figure 6 were performed in slices from Nlg1 WT mice where endogenous Nlg1 was knocked down by shRNA, and rescued by exogenous Nlg1 mutants. Measurement of the basal AMPA- and NMDA-receptor mediated EPSCs suggest a similar level of expression compared to non-electroporated counterparts (Fig. 6A-E), and reveal strong basal differences between the two Nlg1 point mutants (Fig. 6F). Finally, we expressed optoFGFR1 alone in the Nlg1 WT or KO background, and found significant effects of activating the optoFGFR1 only when endogenous Nlg1 is present (see accompanying paper Fig. 2). Taken together, we feel that all these experiments point to the same concept that tyrosine phosphorylation of endogenous Nlg1 is important for synapse differentiation and potentiation.

1) I find it surprising that Nlg1 Y782A has no additional effect over WT Nlg1 in increasing excitatory synapse number, and that Nlg1 Y782F has no additional effect over WT Nlg1 in increasing inhibitory synapses. If the model is that there is normally a (small) pool of WT Nlg1 that is unphosphorylated at Y782 and assembles an inhibitory scaffold via gephyrin, then the Y782A mutant (equivalent to pY782 and incapable of binding gephyrin) would be expected to generate more excitatory synapses than WT. This is not the case in the experiments analysing GluA1 cluster density and mEPSCs, suggesting that WT Nlg1 is already maximally phosphorylated at Y782A. However, if WT Nlg1 is normally maximally phosphorylated at Y782A, then how can it be explained that the Y782F gephyrin-binding mutant does not cause a greater increase in inhibitory synapse formation than WT Nlg1? This suggests to me that describing Y782 as a ‘switch’ is too strong, and that there might be additional residues in the Nlg1 C-tail or gephyrin-independent pathways. The approach of overexpressing WT and Nlg1 Y782 mutants in a WT background might cause heterodimerization with endogenous Nlg. These points should at least be discussed, but any experimental insight would strengthen the paper.

We thank the reviewer for raising this interesting question. We agree with the limits of over-expressing Nlg1 Y782A/F mutants in a wild type background. In fact, when Nlg1 is overexpressed, we think that both the phosphorylated and unphosphorylated forms of Nlg1 become equally abundant, and might occupy all available scaffolding molecules PSD-95 and gephyrin, respectively. As a result, Nlg1-WT apparently induces a maximal effect on both excitatory and inhibitory synapses, which are independently mimicked by the 2 mutants (Y782A and Y782F respectively). In a first approach to address this issue, instead of pooling the data from different cells, we computed the ratio of the AMPA- versus GABA_A-receptor mediated mEPSCs frequency from the same neuron expressing Nlg1-WT, -Y782A, or -Y782F (as suggested in minor point 3). We could do that because the recordings of excitatory and inhibitory miniature currents were made sequentially on the same cells, upon changing the holding potential (from -70 mV to +10 mV). When doing this statistical paired analysis, it becomes more obvious that the effects of Nlg1-WT on excitatory and inhibitory currents fall between those of Nlg1-Y782A and -Y782F, respectively (Fig. 4 below). These effects might have been obscured by the large variability in the miniature post-synaptic current frequencies in the unpaired analysis. This graph is now presented in Fig. S7A and commented in the results.

Figure 4: Excitation to inhibition balance in cultured hippocampal neurons expressing the Nlg1 point mutants. The graph represents the ratio of paired mEPSC to mIPSC frequencies computed on a per cell basis. mEPSCs and mIPSCs were sequentially recorded at -70 mV and +10 mV, respectively, in normal aCSF containing 0.5 μ M TTX. Data were compared by a Kruskal-Wallis test followed by Dunn's multiple comparison test (*P < 0.05, **P < 0.01).

Second, in a new set of experiments dedicated to measure LTP (**Fig. 6**), we expressed the Nlg1 point mutants (Y782A and Y782F) or Nlg1-WT in CA1 neurons, in a near-replacement configuration. Since LTP is reduced or absent in Nlg1 KO mice (see for instance Blundel et al., J Neurosci 2010; Jiang et al., Mol Psych 2016), we used organotypic hippocampal slices from Nlg1 WT mice, so that control unelectroporated neurons are expected to show normal LTP. To reduce the effects of Nlg1 over-expression that might occlude LTP, we co-electroporated CA1 neurons with AP-tagged Nlg1-WT, -Y782A, or -Y782F with shRNA to Nlg1 containing a GFP reporter, and BirA for subsequent labeling with streptavidin. Aside from the LTP responses, CA1 neurons expressing Nlg1 WT displayed basal AMPAR-mediated and NMDAR-mediated EPSCs that were in the same range as those of non-electroporated neurons having endogenous Nlg1, indicating functional rescue (**Fig. 6A-E**). In contrast, neurons expressing Nlg1-Y782A showed 4-fold higher basal EPSCs than non-electroporated counterparts, while neurons expressing Nlg1-Y782F exhibited similar EPSCs as non-electroporated counterparts, despite the fact that the 2 mutants were expressed to similar levels as Nlg1 WT (**Fig. 6F**). Thus, when expression was kept close to endogenous levels, major differences appeared between the two mutants. These data thereby support the concept that Nlg1 can exist in both phosphorylated and unphosphorylated forms, that are mimicked by the Y782A and Y782F mutants. Regarding the phosphotyrosine level of endogenous Nlg1, please see our answer to point 2.

According to the reviewer's remark, we overall moderated the Y782 phosphorylation switch concept throughout the text (and title), insisting more on the fact that NLG1 tyrosine phosphorylation is involved in excitatory synapse differentiation and potentiation.

2) Relevance of Nlg1 Y782 phosphorylation in E/I balance: can the authors provide insight into the phosphorylation state of endogenous Nlg1; does this change over the course of neuronal development and upon manipulation of activity levels? Does increased excitation lead to a Nlg1 dephosphorylation and a shift from excitatory to inhibitory synapses?

To address this comment (related to point 2 raised by Reviewer 1), we examined the regulation of the endogenous level of Nlg1 tyrosine phosphorylation in cultured neurons, by performing Nlg1 immunoprecipitation followed by phosphotyrosine (pTyr) immunoblots. We first examined the Nlg1 pTyr level at different time points in culture (DIV 7, 10, 14, and 17) (**Fig. S8A,B**). The P-Tyr level, normalized by the level of Nlg1, seems higher around DIV 10, a period of active synaptogenesis. This suggests that a signal originating during synaptogenesis is triggering Nlg1 phosphorylation, e.g. the binding to NRXs upon the formation of axon/dendrite contacts. In support of this concept, the tyrosine kinases TrkB and Trk C were found in a proteomics screen of pan-NRX interactors (**not shown**). Second, we tested the effect of several tyrosine kinase inhibitors on the pTyr level of endogenous Nlg1 in neurons, after validating the activity of those inhibitors in COS cells co-expressing Nlg1 and recombinant tyrosine kinases (FGFR1, TrkB, TrkC) (**Fig. 8C,D**). Those tyrosine kinase candidates were chosen from an initial screen relying on an in vitro kinase assay (**Fig. S8C,D and Fig. 8A,B**). We show that, although most inhibitors reduce Nlg1 pTyr levels in neurons to some degree (15-50%), the pan-Trk inhibitor (GNF5823) almost entirely blocks the Nlg1 pTyr level (**Fig. 8E,G**). Encouraged by this result, we tested the effect of GNF 5823 on synaptic transmission in organotypic hippocampal slices. The inhibitor totally blocked the increase of AMPAR-mediated EPSCs caused by the expression of Nlg1-WT in CA1 neurons from the Nlg1 KO background (**Fig. 8H,I**). In contrast, the FGFR1 inhibitor caused a modest reduction of Nlg1 P-Tyr level, and did not prevent the increase in synaptic transmission caused by Nlg1 expression (**Fig. S10**). Together, these new results indicate that the Nlg1 tyrosine phosphorylation level in neurons is significant and can be regulated over developmental age, most likely through the Trk family of tyrosine kinases. In parallel, the optoFGFR1 experiments suggest that endogenous Nlg1 is not maximally (constitutively) phosphorylated, because we can induce major effects on dendritic spines and AMPAR currents by triggering its phosphorylation (**see accompanying manuscript**). Therefore, our conclusion from these experiments is that Nlg1 PTyr level is regulated in neurons, most likely by a balance between phosphorylation via tyrosine kinases such as Trks, and dephosphorylation by phosphatases which remain to be identified. This balance finely regulates the amount of AMPARs that can be recruited by the associated PSD scaffold at excitatory synapses.

Regarding the effects of neuronal activity on NLG1 phosphorylation, we think this is highly relevant to threonine phosphorylation of NLG1 by CamKII, which is enhanced by NMDA receptor activity (Bemben et al., Nat Neurosci 2014). Although our new data suggests an important role for Nlg1-Y782 phosphorylation in activity-dependent processes such as LTP through an unsilencing mechanism, in our view the induction of tyrosine kinase activity controlling Nlg1 phosphorylation might primarily depend on ligand binding (see Giannone et al., Cell reports, 2013). In addition, our initial attempts to stimulate NLG1 tyrosine phosphorylation

upon chemical LTP in primary cultures were not very reliable (data not shown). So we are sorry that we left this suggestion aside for now and concentrated more on exploring the kinase pathway that regulates NLG1 tyrosine phosphorylation.

3) What is the effect of the Nlg1 $\Delta 5$ and $\Delta 72$ on inhibitory synapse formation? The $\Delta 5$ mutant lacks the PSD-95 binding site but can still interact with gephyrin. Does a Nlg1 Y782F/ $\Delta 5$ double mutant increase inhibitory synapse formation over WT Nlg1 levels? In Fig. 6, the Nlg1 $\Delta 5$ and $\Delta 72$ mutants significantly increase AMPAR EPSCs compared to control despite inability to engage with the postsynaptic density scaffold. As these experiments were done in Nlg1 KO slices (correct?), does this suggest heterodimerization with Nlg3, or PDZ domain-independent interactions?

In response to the reviewer's comment, we have performed electrophysiological recordings of inhibitory synaptic transmission in CA1 neurons expressing Nlg1-WT, -Y782A, or -Y782F slices in organotypic slices from Nlg1 KO mice. Surprisingly, all Nlg1 constructs induced a similar ~5-fold enhancement of IPSCs compared to non-electroporated neighboring neurons, independently of the mutation, while we were expecting Nlg1-Y782A to prevent this increase (Fig. S7B,C). Taken together, these results indicate that the Nlg1 tyrosine point mutants act differently on inhibitory synapse differentiation in organotypic slices and in dissociated cultures. Thus, these experiments did not encourage us to assess the effects of the other mutants Nlg1 $\Delta 5$ and $\Delta 72$ or NLG1 $\Delta 5$ -Y782F as suggested by the reviewer, on inhibitory synaptic transmission.

Yes, the electrophysiological recordings in slices were done in the Nlg1 KO background. Heterodimerization with NLG3 is suspected and mentioned in the discussion.

4) The Nlg1 Y782F mutant is described as 'blocking' excitatory synapse formation when overexpressed in WT cultured neurons (pg. 8), but when expressed in KO slices still doubles AMPAR EPSC amplitudes compared to controls (Fig. 5C, not significant here). The description of Y782F as 'blocker' is too strong in my opinion. How do the authors explain these different observations?

The reviewer is right that the effects of the NLG1 Y782F mutant depend on the experiment performed. In primary cultures, regarding excitatory synapses, the Y782F mutant significantly decreases GluA1 accumulation increases its lateral mobility, and prevents the increase in mEPSC frequency (Figs 2 and 3). Regarding inhibitory synapses, Y782F increases $\gamma 2$ accumulation, decreases its mobility, and increases mIPSC frequency (Figs 7 and S6). However, as noticed by the reviewer, Nlg1-Y782F still increases by 2-fold AMPAR-mediated EPSCs in organotypic hippocampal slices from Nlg1 KO mice, potentially by heterodimerizing with NLG3. However, it is unclear why Nlg1-Y782F has such a strong effect in primary neurons where it may also homodimerize with endogenous Nlg3. One explanation for the difference is perhaps the over-expression level between the two systems. Anyway, we moderate our sentences, and mentioned this potential Nlg1/3 heterodimerization mechanism.

5) The surface expression levels of Y782F are lower than Y782A based on Nrx binding (Fig. S2). The decreased colocalization of the Y782F mutant with PSD-95 shows a small decrease that could also be explained by lower surface expression of the mutant. Please comment.

We think that the difference between the 3 conditions (statistically not significant) arose from the small number of Nlg1-Y782F expressing neurons that were detected in this particular experiment. Our previous study with the same mutants indicated no significant differences between the surface expression levels of Nlg1-WT, Y782A, and Y782F, as labeled with anti-HA or Nrx1 β -Fc (Giannone et al., Cell Reports 2013, FigS4). Moreover, expressing the Nlg1-Y782F mutant in CA1 cells from organotypic slices increases NMDAR-EPSC amplitude to the same extent as Nlg1-WT and Nlg1-Y782A. Given that NMDARs and Nlg1 likely interact through extracellular coupling (Budreck et al., PNAS 2013), this further suggests that all three constructs express similarly at the cell membrane.

6) What is the effect on PSD-95 binding of Y782A and Y782F? This can be easily tested using the C-tail constructs in Fig. 1C.

We now provide in Fig. 1D an anti-PSD95 immunoblot showing the pull-down of recombinant PSD-95-mcherry by the three GST-Nlg1 proteins, and corresponding quantification. As expected from identical sequences in

their PDZ domain-binding motifs, the levels of bound PSD-95 are similar for Nlg1-WT, -Y782A and -Y782F, while GST alone does not bind PSD-95. Thus, *in vitro* the Y782 mutations affect essentially gephyrin binding.

7) The section describing the Nlg3 mutants is not clear to me. Did the authors test the effect of the Nlg3 mutations on gephyrin binding? This can be easily tested.

In response to the reviewer's remark, we reorganized the main text to include a paragraph at the end of the results entitled "Regulation of synapse differentiation by tyrosine phosphorylation is specific to Nlg1". In this paragraph, we pooled our additional results regarding phosphorylation of Nlg2 and Nlg3 with the initial results regarding the Nlg3 point mutants (Y792A/F). We hope that it reads better now.

When expressed in COS cells, both Nlg2 and Nlg3 were phosphorylated by recombinant tyrosine kinases such as TrkB/C (Fig. S8E). In neurons however, only Nlg1 and Nlg3, but not Nlg2, were tyrosine phosphorylated, potentially reflecting the preferential presence of endogenous tyrosine kinases such as Trks at excitatory synapses. Regarding Nlg3, the Nlg3 phosphorylation signal in neurons was only partially blocked by the pan-Trk inhibitor, suggesting that other kinases might phosphorylate an additional tyrosine residue specific to Nlg3 (Y827). Indeed, a Nlg3 construct bearing only the Y792A mutation was still significantly phosphorylated in COS-7 cells compared to Nlg3-WT (Fig. S4A-C). Since Nlg3 is also present at glutamatergic synapses, we investigated whether mutations of the conserved tyrosine residue in Nlg3 (Y792) affected miniature AMPA currents (Fig. S4A). The two Nlg3 phosphomutants (Y792F and Y792A), as well as Nlg3-WT, equally increased the frequency of AMPAR-mediated mEPSCs compared to neurons expressing empty vector (Fig. S4D-E). Taken together, these experiments lead us to conclude that the regulation of glutamatergic synapse differentiation by Nlg tyrosine phosphorylation is specific to the Nlg1 isoform. Finally, the fact that activation of the optoFGFR1 does not affect inhibitory currents, argues against a major role of Nlg2/3 phosphorylation on inhibitory synaptic transmission (accompanying manuscript Fig. S1). Thus, we did not feel imperative to measure the binding of Nlg3 Y792A/F mutants to gephyrin.

Minor comments:

1) In the intro it is mentioned there are 3 Nlgs in rodents, this should be 4: Nlg1-4.
This was corrected.

2) What is the evidence for endogenous Nlg1 at inhibitory synapses? The authors mention a 'sizeable fraction' (pg. 12) at inhibitory synapses and cite two papers (refs 11 and 29), but ref 11 only shows exogenously expressed tagged Nlg1 at inhibitory synapses and ref 29 relies on biotinylating probes fuse to exogenously expressed synaptic adhesion proteins. Is there any evidence for endogenous Nlg1 at inhibitory synapses?

We appreciate the reviewer's remark and have now removed this sentence. We have slightly shifted the initial focus of the paper to better highlight the role of Nlg1 tyrosine phosphorylation in excitatory synapse differentiation. Our new results showing bidirectional and time-dependent regulation of NGL1 tyrosine phosphorylation argue that one mechanism by which Nlg1 is selectively retained at excitatory synapses indeed involves tyrosine phosphorylation by TRKs, which inhibits Nlg1 binding to gephyrin and thus makes it unable to form inhibitory synapses.

3) Although the authors measure E and I synaptic transmission here in separate experiments (Fig. 2/ Fig. 4); it would be interesting to show E/I ratios of transmission properties measured from the same cell, both in cultured neurons as in the organotypic slices and determine how E/I balance is affected by the different Nlg1 mutants.

Please see our response to major point 1. In cultured neurons, we computed the ratio between mEPSC and mIPSC frequency for each cell, and found more striking differences between Nlg1 WT, Y782A, and Y782F than when cells were pooled and compared by unpaired design (Fig. S7A). In slices, we did not perform the recordings of excitatory and inhibitory transmission on the same cells, so such a paired analysis is not possible.

4) Please change in text of first results paragraph: "...full length Gephyrin to entire Nlg1", change to C-tail of Nlg1.
Done.

5) Please indicate n numbers in figure legends.

The n numbers are either given within the graph bars, or represented as individual data points.

6) 1E, G: Please add color code to text left of each panel (B, R, G), this will help to interpret the merged crop for each condition represented.

Done.

7) Pg. 10 2nd paragraph: reference to figures with spine density data is wrong: change 5F to 5H; change 5F, G to 5H, I.

Done.

8) Pg. 10 3rd paragraph: add reference to commercial screen for tyrosine kinase or show results in supplementary data.

The commercial screen was performed by Prokinase GmbH (Germany), and involved from our side the production of biotinylated Nlg1 peptides comprising the 16-aa gephyrin-binding motif that were already described (Giannone et al., Cell Rep 2013). The company screened 81 tyrosine kinases in their ability to phosphorylate those peptides immobilized on streptavidin-coated wells, by detection of bound radioactive phosphate. We now show the results of this screen in **Fig. S8C** and give details in the methods section. However, these data should be taken only as indicative, because some of the tyrosine kinases like Tyro3 were not found in a second smaller screen restricted to the strongest candidates (not shown). This is why we generated GST-NLG1 proteins and tested ourselves their ability to be phosphorylated by recombinant tyrosine kinases.

9) The authors should cite Nguyen et al., 2016 eLife, who identify gephyrin-dependent and -independent pathways in the Nlg2 C-tail.

This paper is now cited in the discussion.

REVIEWERS' COMMENTS:

Reviewer #2 (Remarks to the Author):

The authors have addressed all previous concerns/comments. This paper convincingly demonstrates that the phosphostate of Nlg1 Y782 can dictate its role at excitatory and inhibitory synapses.

Reviewer #3 (Remarks to the Author):

The authors have made considerable effort to revise their manuscript and address my concerns. The authors changed the focus of the study and added more relevant data showing that endogenous Y782 phosphorylation of Nlg1 is important for excitatory synapse differentiation and long-term potentiation. The reorganised text and figures flow better and I agree with the decision to leave out the optoFGFR data (although the observation that this only has an effect in the presence of endogenous Nlg1 is compelling and supports their model). The manuscript is suitable for publication in Nat Comms. Upon reading the revised manuscript, a few minor questions arose. I would appreciate it if the authors could address/correct these:

- pg 5 lines 120-124: Differential diffusion coefficient of two mutants compared to WT. Y782A is slower, Y782F is faster, this is correlated with their different binding properties to PSD/Gephyrin scaffolds but why would Nlg1 associated more to Gephyrin be more mobile? Is this because they are not trapped in a spine? Or because Nlg1 at Gephyrin-sites interacts less with presynaptic partners? Please clarify.
- pg 7 line 198 Figure reference is wrong, please change 4H, I to Fig. 4F,G. Similarly, adjust correct figure references on pg8 line 231, 233 and 235 to 4H,I.
- pg 9 line 285 please add behind (mIPSCs): 'in primary hippocampal cultures' to make better contrast with slices that are described below.
- Rename S10 and S9, they have been switched in the figures
- Fig. S9: only 1 spectral count questions the significance of the pull-down of the Trk kinases in this assay. In addition, this affinity purification is done with antibodies for presynaptic neurexins, whereas the model suggests a cis-interaction with Nlg1. I don't find this data particularly convincing or supporting the model. I strongly suggest to leave this out. Detection of Trks in Nlg1 IP with western or mass spectrometry would be convincing evidence, but such an interaction might be low-affinity or transient and therefore difficult to detect.
- Model in Fig. 9: A silent synapse can also be at excitatory immature spines where Nlg1 associates with PSD and NMDA but has not recruited AMPAR. The depiction of the 'hybrid' excitatory NMDAR-containing synapse with inhibitory scaffold and GABAR on the dendritic

shaft as 'silent synapse' is unconventional. To what extent such synapses exist is unclear (although discussed in the Discussion). I would change this cartoon to more clearly include the effect of -A (resembles high phosphorylation state) versus -F mutant (resembles non phosphorylated state) in the summary scheme instead.

Response to reviewers

Reviewer #3 (Remarks to the Author):

The authors have made considerable effort to revise their manuscript and address my concerns. The authors changed the focus of the study and added more relevant data showing that endogenous Y782 phosphorylation of Nlg1 is important for excitatory synapse differentiation and long-term potentiation. The reorganised text and figures flow better and I agree with the decision to leave out the optoFGFR data (although the observation that this only has an effect in the presence of endogenous Nlg1 is compelling and supports their model). The manuscript is suitable for publication in Nat Comms. Upon reading the revised manuscript, a few minor questions arose. I would appreciate it if the authors could address/correct these:

We are glad that this reviewer acknowledges our efforts to improve the manuscript, and for accepting to leave out the optoFGFR1 data for another manuscript.

- pg 5 lines 120-124: Differential diffusion coefficient of two mutants compared to WT. Y782A is slower, Y782F is faster, this is correlated with their different binding properties to PSD/Gephyrin scaffolds but why would Nlg1 associated more to Gephyrin be more mobile? Is this because they are not trapped in a spine? Or because Nlg1 at Gephyrin-sites interacts less with presynaptic partners? Please clarify.

We thank the reviewer for this insightful comment. In fact the Nlg1-Y782A diffuses slightly less fast than Nlg1-WT (the difference is not significant), probably because they are similarly trapped at PSD-95 scaffolds. The biggest difference is observed with Nlg1-Y782F, whose global diffusion is higher than Nlg1-WT and -Y782A. The fraction of Nlg1-Y782F which is associated to gephyrin scaffolds is probably as confined as Nlg1-WT and Y782A at PSD-95 scaffolds, but the fact that gephyrin puncta are relatively less numerous than PSD-95 puncta (20 % vs 60 % for this mutant, see Figs. 1f;h) leaves more room for Nlg1-Y782F extra-scaffold diffusion. We added one sentence in the results section to clarify this point.

- pg 7 line 198 Figure reference is wrong, please change 4H, I to Fig. 4F,G. Similarly, adjust correct figure references on pg8 line 231, 233 and 235 to 4H,I.

Done.

- pg 9 line 285 please add behind (mIPSCs): 'in primary hippocampal cultures' to make better contrast with slices that are described below.

Done.

- Rename S10 and S9, they have been switched in the figures.

Done

- Fig. S9: only 1 spectral count questions the significance of the pull-down of the Trk kinases in this assay. In addition, this affinity purification is done with antibodies for presynaptic neurexins, whereas the model suggests a cis-interaction with Nlg1. I don't find this data particularly convincing or supporting the model. I strongly suggest to leave this out. Detection of Trks in Nlg1 IP with western or mass spectrometry would be convincing evidence, but such an interaction might be low-affinity or transient and therefore difficult to detect.

We agree with the reviewer's remark that the proteomics data did not offer very strong evidence in support of the model, due to the small peptide counts for Trks. However, we think that those kinases do not need to be in very high amounts to phosphorylate Nlg1, e.g. in the in vitro kinase assay (Fig. 8a,b), trace amounts of recombinant kinases are enough to strongly phosphorylate Nlg1. So overall we do not expect to detect major amounts of tyrosine kinase receptors associated with the Nrx-Nlg complex. The reviewer is also right that the interaction between Nlg1 and the RTK is probably very labile, as we did not detect Trks in a proteomics screen from a Nlg1 IP (not shown). We suspect that the RTK is more likely to be associated with Nrxs in a trans-synaptic fashion, although at this stage we do not have a direct proof of this mechanism. So, we are following the reviewer's advice and leave out the proteomics data and methods from this manuscript. As a consequence, we also removed the name of our collaborator who performed this analysis (Jeffery Savas, Chicago) from the author list.

- Model in Fig. 9: A silent synapse can also be at excitatory immature spines where Nlg1 associates with PSD and NMDA but has not recruited AMPAR. The depiction of the 'hybrid' excitatory NMDAR-containing synapse with inhibitory scaffold and GABAR on the dendritic shaft as 'silent synapse' is unconventional. To what extent such synapses exist is unclear (although discussed in the Discussion). I would change this cartoon to more clearly include the effect of -A (resembles high phosphorylation state) versus -F mutant (resembles non phosphorylated state) in the summary scheme instead.

We thank the reviewer for this insightful comment. We admit that our cartoon of a silent synapse with both GABA-A and NMDA receptors was a little unusual. We refined it to separate the two observed phenotypes for Nlg1-Y782F: 1) a classical silent synapse with NMDA receptors and no AMPA receptors (no RTK either and weak PSD scaffold), associated with a glutamatergic axon; and 2) Nlg1-Y782F associated with gephyrin scaffolds and GABA-A receptors, in front of a GABAergic axon. Both types of synapses are on the shaft, i.e. not forming dendritic spines. In contrast, Nlg1-Y782A mimics Nlg1 phosphorylated by RTKs: it potentiates excitatory synapses, forming dendritic spines containing both AMPARs and NMDARs. We hope that this modified cartoon will be more convincing.